# A single-cell and spatial atlas of early human olfactory development

Yvon Mbouamboua[1,2,3], Kevin Lebrigand [2,3], Sreekala Nampoothiri [1], Marie Couralet [2], Marie-Jeanne Arguel [2], Ludovica Cotellessa [1], Cécile Allet [1], Vincent Prevot [1], Pascal Barbry [2,3] ✉ & Paolo Giacobini [1,3] ✉

The human nasal region arises from neural crest and placodal lineages, yet its early development remains poorly understood owing to limited fetal tissue access and structural complexity. Here we present an integrated single-nucleus and spatial transcriptomic atlas of the human fetal nasal region, generated from male and female fetuses between 7 and 12 post-conceptional weeks. Single-nucleus RNA sequencing [snRNA-seq] resolved 32 distinct cell types, while integration with multiplexed error-robust fluorescence in situ hybridization (MERFISH) enabled spatial and temporal mapping of gene expression dynamics across the olfactory epithelium (OE) and adjacent tissues. We identify markers of olfactory sensory neuron differentiation and pathways governing epithelial patterning and OE morphogenesis. Notably, spatially resolved snRNA-seq profiles of 169 olfactory receptor genes reveal molecular support for the "one neuron-one receptor" principle already in the first trimester. Together, this work establishes a molecular and spatial framework of early human olfactory development and provides a resource for studies of sensory neurogenesis and congenital disorders.

The sense of smell, operated by the human olfactory system, is a major component of our outside perception that contributes to our interactions with complex environments. It plays a crucial role in vital functions such as feeding, ability to detect hazardous odors, and social relationships. Olfactory loss can profoundly affect a person's quality of life and it can lead to reduced appetite, poor nutrition and even neurological disturbances such as anxiety or depression[1,2].

Olfactory sensory neurons (OSNs), the primary detectors within the main olfactory epithelium, have evolved under strong selective pressure to recognize an odorant chemical space of more than 10,000 distinct moelcules[3]. Odorant molecules are detected by a family of transmembrane receptors, related to the G protein-coupled receptors, in which they represent more than 50% of the total number of functional genes, with over 1200 olfactory receptor (OR) genes in mice and approximately 400 in humans[4,5]. Each OSN expresses a single OR gene in a monogenic and monoallelic manner[6–8]. Commitment to a given OR is governed by multiple mechanisms. H enhancers[9,10], for instance, act in mouse as multi-chromosomal enhancer hubs that activate only one OR locus[11,12], and a feedback loop dependent on OR protein translation[13,14]. Recent research also points to a RNA-mediated symmetry-breaking mechanism that reinforces singular OR gene expression[15].

Single-cell transcriptomic studies in postnatal rodents have further delineated distinct molecular stages of OSN differentiation and dynamic shifts in OR gene expression. Notably, previous work showed that in mice, mature OSNs adopt a single-OR expression profile through a developmental sequence that appears independent of sensory activity[8]. Whether this developmental process is conserved in humans, however, remains an open question.

The mouse OE appears around embryonic day 9 (E9) with the formation of the olfactory placode, a specialized ectodermal structure at the anterior end of the embryo. This placode invaginates to form the

[1]Univ. Lille, Inserm, CHU Lille, Lille Neuroscience & Cognition, UMR-S 1172, Lille, France. [2]Université Côte d'Azur, CNRS, INSERM, Institut de Pharmacologie Moléculaire et Cellulaire, IHU RespirERA, 3IA Côte d'Azur, Sophia Antipolis, France. [3]These authors contributed equally: Yvon Mbouamboua, Kevin Lebrigand, Pascal Barbry, Paolo Giacobini. ✉e-mail: barbry@ipmc.cnrs.fr; paolo.giacobini@inserm.fr

olfactory pit, giving rise to the olfactory epithelium. As development progresses, this epithelium becomes regionally patterned through coordinated mesenchymal-epithelial interactions. OSNs begin differentiating by mid-gestation (E11–E13), and their axons project toward the developing olfactory bulb. By late gestation, the OE contains distinct zones with differentiated OSNs, supporting cells, and progenitor populations, reflecting the establishment of a functional olfactory system well ahead of birth[16–18].

Studies in model organisms, particularly mice, have been instrumental in defining OSN specification, epithelial patterning, and OR gene regulation during development. However, key species-specific differences constrain the direct translation of these insights to human olfactory biology.

We took advantage of recent advances in single-cell transcriptomics and spatial multi-omics to profile human nasal region during post-conceptional weeks (PCW) 7–12 at high resolution, capturing not only the identities and states of individual cells but also their spatial relationships and lineage histories. We constructed a high-resolution developmental cell atlas of the human nasal region between PCW7 to PCW12 by integrating single-nucleus RNA sequencing (snRNA-seq) and multiplexed error-robust fluorescence in situ hybridization (MERFISH)[19]. snRNA-seq profiling of 41,875 nuclei identified 32 major cell types involved in the formation of the nasal region. Our analysis reveals the molecular programs driving cell differentiation, spatial patterning, and lineage trajectories across epithelial, cartilaginous, immune, neuronal, glial, muscular, and vascular compartments. In complement, MERFISH profiling of 188,803 cells uncovers insights into the localization and interactions of progenitor populations that shape the olfactory system architecture.

A major finding of our study is the characterization of human OSN development, with an already strong expression of some specific human ORs during the first trimester. Our data confirm that most human OSNs, even at these early stages of development, already express a unique OR. These findings provide insights into the early molecular events governing olfactory function.

Together, our work provides a reference for understanding the cellular and molecular architecture of the developing human olfactory system. It lays groundwork for future studies of craniofacial and sensory disorders, and for exploring the mechanisms underlying congenital anosmias and other developmental anomalies of the nasal region.

## Results

### Single-nucleus atlas of the human fetal nasal compartment

To characterize the cellular landscape of the developing human nasal region, we performed snRNA-seq on eight fetal samples spanning PCW7-PCW12 (Fig. 1a). To mitigate limited tissue availability and reduce experimental variability, samples were pooled and nuclei were reassigned to donors *post hoc* using genotype-based demultiplexing[20]. Cryosection pooling enabled efficient isolation of high-quality nuclei, which were profiled using a 10x Genomics snRNA-seq platform. Donor identities were resolved using Demuxafy, combining Souporcell and Vireo for robust genotype-based assignment without requiring prior genetic information[21,22] (Supplementary Fig. 1a).

Bulk RNA-seq genotypes showed high within-donor concordance, enabling clear donor separation and reliable genotype calling (Supplementary Fig. 1b). After filtering variants with >20% missingness, donor call rates exceeded 95%, yielding a high-quality SNP set suitable for demultiplexing. Single-cell donor assignments across two independent sequencing runs closely matched bulk genotypes, with Vireo and Souporcell accurately assigning most singlets and minimal cross-donor misassignment (Supplementary Fig. 1c, d, Supplementary Data 1).

Following sample demultiplexing and donor assignment of individual nuclei (Fig. 1b), datasets from all developmental stages were integrated to generate a unified transcriptional map of 41,875 nuclei (Fig. 1c, Supplementary Data 2). The majority of nuclei were classified as singlets, with smaller fractions identified as doublets or unassigned (Supplementary Fig. 2a). After quality-control filtering (Supplementary Fig. 2b–d), a balanced, high-quality set of donor-matched nuclei was retained for downstream analyses (Supplementary Fig. 2e).

Sex was assigned using per-nucleus XIST-to Y-linked gene expression ratios, classifying nuclei as male-like, female-like, or ambiguous (Supplementary Fig. 3a, b; Supplementary Data 3). A small subset with discordant sex signatures and weak erythroid gene expression, consistent with minimal maternal blood-derived ambient RNA, was removed during quality control (Supplementary Fig. 3c, d; Supplementary Data 3). Low erythroid scores across samples (≤2%) supported minimal contamination and yielded a sex- and donor-consistent dataset (Supplementary Fig. 3e).

Hierarchical clustering of gene expression correlations resolved the fetal nasal region into major cellular compartments, including epithelial, neuronal, mesenchymal, vascular, immune, muscle, and neural crest-derived glial populations (Supplementary Fig. 4a–d, Supplementary Fig. 5a; Supplementary Data 4, 5).

To assess temporal changes in cellular composition, we quantified cell type proportions across developmental stages (Supplementary Fig. 5b; Supplementary Data 2). Mesenchymal compartments showed pronounced dynamics, with multiple mesenchymal and osteogenic subtypes increasing over time, consistent with active craniofacial growth and tissue remodeling, whereas neural crest cells (NCCs), GnRH neurons, and NOS1 neurons exhibited an inverse trend.

GnRH neurons, which regulate mammalian fertility, originate in the olfactory placode and migrate into the brain during early fetal development along the terminal and vomeronasal nerves[23–25]. At PCW6-7, these neurons migrate through the nasal region together with placode- and neural crest-derived cells of the migratory mass (MM)[25,26]. Consistent with this process, we observed a pronounced reduction in GnRH neurons and NOS1+ MM components in the nasal region between PCW7 and PCW12, reflecting their progressive relocation into the brain (Supplementary Fig. 5b). UMAP embeddings at two different levels of annotation show a coherent organization of the different cellular entities (Supplementary Fig. 5c, d).

The resulting unsupervised clustering and annotation identified 32 distinct cell populations (Fig. 1d), that were consistent with previous studies in other models[8,18,25–27]. Neuronal and glial populations displayed a large transcriptional diversity, with olfactory neurons, neural progenitors, neural crest cells and their derivatives, but also Schwann cells and olfactory ensheathing cells (OECs) (Fig. 1d, e). Non-neuronal populations comprised respiratory epithelial cells, mesenchymal stromal cells, pericytes, cartilage and osteogenic lineages, endothelial cells, skeletal muscle, and satellite cells (Fig. 1d, e). We also identified abundant immune cell populations, suggesting a potential link between the nasal region and fetal immune development.

No sex-specific differences in cell-type composition were observed across developmental stages (Fig. 1e). Quality-control metrics confirmed the high quality of the snRNA-seq dataset (Fig. 1f).

We next visualized marker gene expression across grouped functional categories. Distinct gene signatures were observed for epithelial, neural crest derivatives and neurons, mesenchymal and cartilagenous populations, vascular, muscle and immune lineages (Fig. 1g; Supplementary Data 6), enabling confident classification of cell identities throughout the developing olfactory system. Many of the top expressed genes represented canonical cell type markers; however, more than half have not previously been reported, providing a valuable resource for future studies (Fig. 1g; Supplementary Data 6).

We assessed cell-cycle dynamics during early human nasal development, in OE, using canonical gene expression signatures. UMAP embedding revealed clear segregation of cells by cell-cycle phase, including G1, S, G2/M, and post-mitotic states (Supplementary

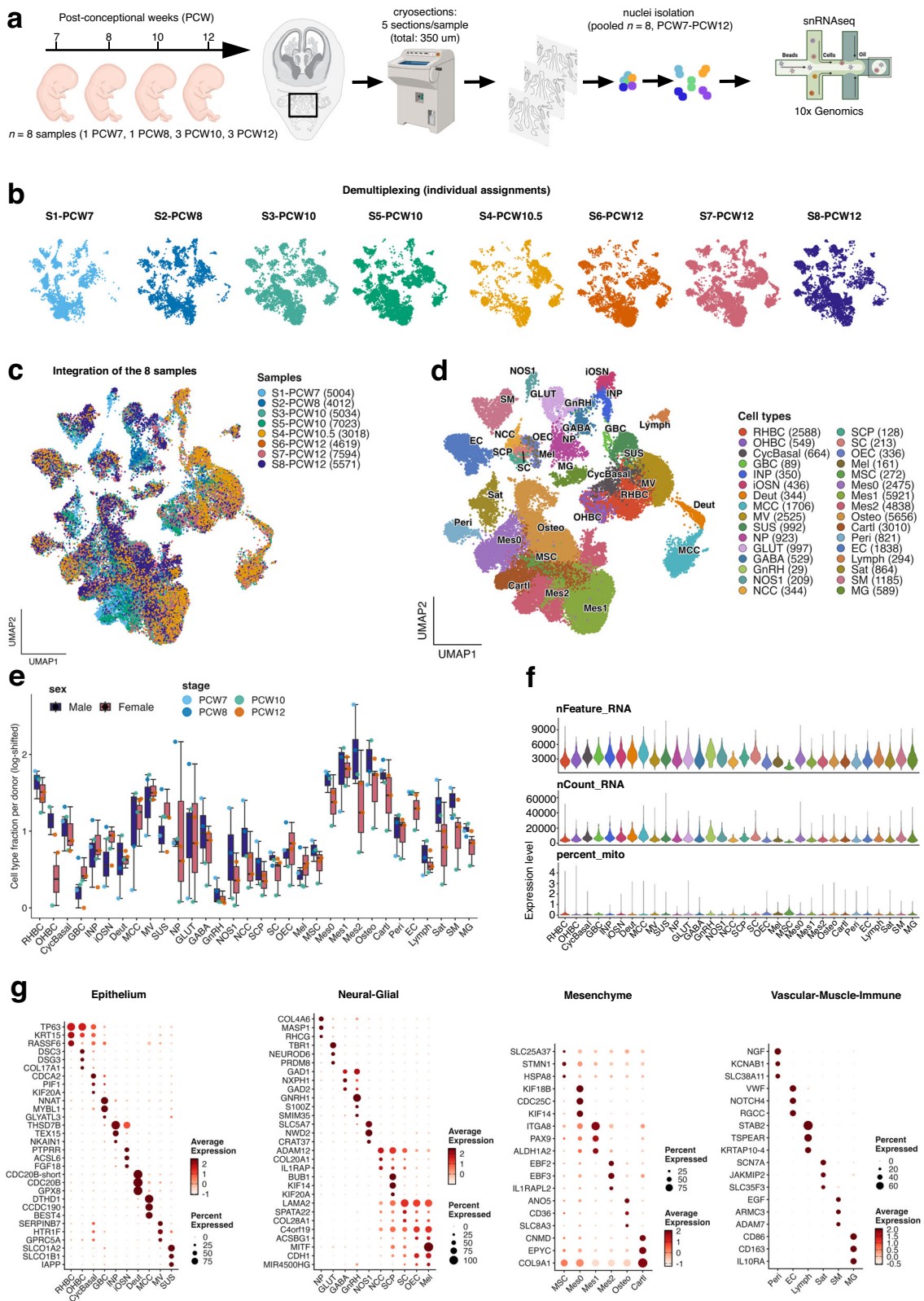

Fig. 6a). Quantification across developmental time showed a progressive decline in actively cycling (G2/M) cells between PCW7 and PCW12, accompanied by an increase in G1 cells, consistent with a transition from proliferation to differentiation (Supplementary Fig. 6b). Cell-type-resolved analysis revealed that cycling basal cells, globose basal cells, Schwann cell precursors, and mesenchymal populations accounted for most S- and G2/M-phase cells, whereas

neuronal populations, including immature OSN (iOSN), immediate neuronal precursors (INP), glutamatergic (GLUT), GABAergic (GABA), GnRH, and NOS1 neurons, were predominantly post-mitotic (Supplementary Fig. 6c). These patterns were corroborated by canonical cell-cycle marker expression and phase scores (Supplementary Fig. 6d).

Together, these data define the temporal and cellular landscape of proliferation in the developing human nasal region and highlight a

**Fig. 1 | Single-nucleus RNA sequencing of the developing human olfactory sensory epithelium. a** Schematic overview of the experimental design and snRNA-seq workflow. Figure created in BioRender, and is licensed under CC BY 4.0 (https://BioRender.com/c1vk104). **b** UMAP projection of demultiplexed nuclei from eight individual donors (S1-S8), colored by sample of origin. **c** Integrated UMAP embedding of all samples (PCW7-12), colored by donor identity. **d** UMAP projection of the integrated dataset colored by annotated cell types. Thirty-two transcriptionally distinct clusters were identified and annotated based on canonical marker gene expression. Cell types include basal, neuronal, epithelial, mesenchymal, vascular, and immune populations. **e** Sample-level cell type fractions stratified by sex. Box plots depict the median, interquartile range (IQR), and whiskers extending to 1.5× the IQR. Individual samples are represented as points, with colors or shapes indicating sex (female, $n = 5$; male, $n = 3$). Fractions were calculated per specimen and log-transformed for visualization. **f** Violin plots showing quality-control metrics across annotated cell types, including the number of detected genes per nucleus (nFeature_RNA), total UMI counts (nCount_RNA), and mitochondrial transcript percentage. Nuclei with fewer than 500 detected genes or UMIs, mitochondrial content >5%, or identified as outliers by a median absolute deviation-based filter were excluded prior to integration. **g** Dot plots of representative marker gene expression across major cell types, grouped by functional categories. Dot size indicates the proportion of cells expressing each gene, and color intensity reflects average scaled expression. Marker genes were identified using two-sided MAST tests with Benjamini-Hochberg correction for multiple testing (padj < 0.05), log-fold change >0.25 and detected in ≥25% of cells in at least one cell type. Data were derived from eight independent samples (PCW7-12). Source are provided as Source Data file.

coordinated shift from progenitor expansion to neuronal differentiation during the first trimester.

## Composition and development of the olfactory system in human fetuses

We next focused our attention to the olfactory system, composed by the olfactory and respiratory epithelium. Cell type identities for each cluster were determined using a combination of well-established murine and human marker genes. We identified canonical OE cell populations, including globose and horizontal basal cells (GBC and OHBC), INP, iOSN, sustentacular cells (SUS), and microvillar cells (MV), alongside adjacent respiratory epithelial populations (Fig. 2a, b)[8,28–30]. Transcriptomic profiling resolved clear molecular distinctions among basal cell states. GBCs expressed progenitor-associated genes such as *MYBL1*, *NNAT*, *HES6*, whereas OHBCs were enriched for *MEG3*, *TP63*, *KRT5* (Fig. 2c, d). Notably, *MEG3*, a long non-coding RNA implicated in epithelial regulation, was enriched in OHBCs compared with RHBCs (Fig. 2c, d), consistent with previous findings in the human adult OE[29].

Quantification of lineage abundance in the OE over developmental time demonstrated a significant decline of CyclBasal, GBC and INP paralleled by an expansion of MV cells and iOSN, mainly from PCW7 to PCW8 (Fig. 2e). OHBC were absent at the earliest developmental time points (PCW7 and PCW8) and they appear at PCW10 and PCW12 (Fig. 2e).

To validate the spatial organization and anatomical context of transcriptionally defined olfactory epithelial cell types, we performed immunofluorescence and RNAscope experiments at PCW9 (Fig. 2f, g). In addition, spatial transcriptomic mapping with MERFISH further validated the anatomical localization of distinct cell populations based on marker genes included in a 294-gene MERFISH panel (Supplementary Fig. 7a–d).

Olig2 has been shown to be transiently expressed in iOSNs and to be downregulated in the mature OSNs in mice from early gestation to adulthood, thus contributing to the final maturation of the OSNs[31]. In agreement with that, at PCW9, OLIG2 immunostaining delineated the OE and distinguished it from the adjacent respiratory epithelium (RE), confirming regional segregation within the developing nasal cavity (Fig. 2f). We also detected *MUC5AC* expression in the RE and the presence of *GNRH1*-positive neurons migrating across the nasal mesenchyme (Fig. 2g).

Transmission electron microscopy of the OE at PCW7.5 revealed its epithelial ultrastructure, showing the presence of MV (pink), SUS (green), OSN (blue) and basal cells (yellow) in the OE and thus providing ultrastructural confirmation of early olfactory epithelial specialization (Fig. 2h).

SOX2, a marker for neural stem cells and progenitors, has been reported to be expressed in olfactory stem cells of the basal OE and RE and in sustentacular cells of the apical OE during late gestation and adulthood in mice[32,33]. Immunostaining for neuronal and progenitor markers revealed laminar organization of the OE, with TUJ1+ neurons in suprabasal layers extending iOSN axons toward the olfactory bulb, and

SOX2+ progenitors localized to basal and intermediate OE layers and throughout the RE (Fig. 2i). Consistent with transcriptomic data (Fig. 2e), OHBC were not detected in the OE at PCW7.5, as indicated by the absence of KRT5 expression, which was restricted to the RE at this stage (Fig. 2i). By PCW12, KRT5+ OHBC were present in both OE and RE, accompanied by marked differences in epithelial thickness and organization (Fig. 2j). Co-staining for SOX2 and KRT5 confirmed basal lamina localization of basal cells in both epithelia and revealed OHBC in the OE arranged in a layered, reactive-like morphology[29] (Fig. 2j).

Together, the molecular annotations derived from transcriptomic data and demonstrate that the human fetal olfactory epithelium exhibits defined regional boundaries, laminar architecture, and neuronal connectivity by the first trimester.

In light of previously documented developmental trajectories that have been mostly achieved in murine OE cells[8,28,34,35], a number of questions can be addressed concerning human OE development and fetal stem cell niche composition. We focused our attention on three olfactory cell populations: GBC, INP, iOSN (Fig. 3a, b). As documented in the adult human OE[29], the fetal GBCs layer was enriched in the expression of *HES6* and with the proliferative markers *TOP2A* (Fig. 3c). Other canonical markers enriched in the human adult GBCs (*NEUROG1* and *NEUROD1*)[29] were detected at low levels in the fetal GBC. Instead, these cells were enriched for transcription factors associated with progenitor proliferation and early neuronal commitment, including *NNAT*, *MYBL1*, and *FOXN4* (Fig. 3c).

INP were characterized by high levels of expression of *TEX15*, *UNC5D*, *ANKFN1*, *FRMPD4*, associated with neuronal maturation and axonal targeting, neuronal commitment and differentiation (Fig. 3c).

We also found in this study that in human fetal olfactory neuroepithelium, *GAP43*, *GNAL*, *ADCY3*, *PTPRR* and *SEMA3E* were expressed in iOSN (Fig. 3c). The expression of these markers in iOSNs indicates that key components of neuronal growth and olfactory signaling are already engaged during early stages of human olfactory neuron differentiation.

Temporal analysis from PCW7 to PCW12 revealed a progressive increase in the abundance of olfactory neuronal populations within the olfactory epithelium (OE) (Fig. 3d). Cell-type identities were assigned using established marker genes[29], resolving OHBC, CyclBasal, GBC, INP, iOSN, MV and SUS cells (Fig. 3e). Markers for OSNs included *GNAL*, *GNG8*, *OLIG2* and *RTP1*[29] (Fig. 3e).

Lineage tracing in mice indicates that OE cells arise from quiescent HBC via proliferative GBC progenitors, which generate both neuronal and non-neuronal lineages[28,35]. Whether similar lineage trajectories operate in humans was unclear. To address this, we applied trajectory inference to human OE snRNA-seq data using Slingshot[36], integrating cluster identities and UMAP embeddings to infer branching lineages (Fig. 3f). This analysis resolved distinct neuronal, microvillar, and sustentacular trajectories emerging from shared progenitors, supported by pseudotime ordering (Fig. 3f, g). Pseudotime density analyses revealed enrichment of neuronal and non-neuronal cells at early developmental stages (PCW7-8), followed by a

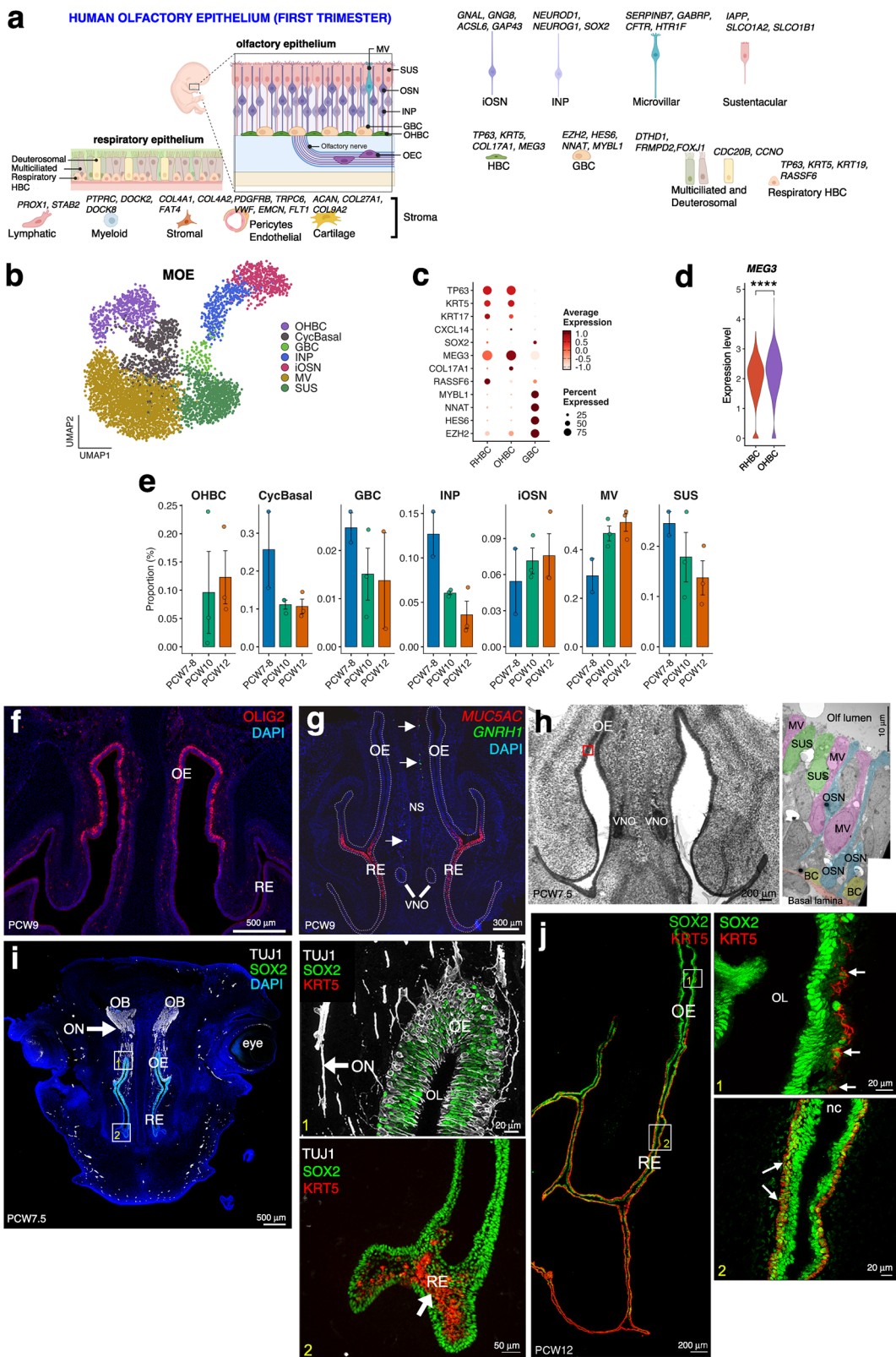

shift toward non-neuronal, particularly microvillar, lineages at later stages (PCW10-12) (Fig. 3h). Together, these data support a model in which GBC act as bipotent progenitors that generate INP and iOSN along the neuronal trajectory, with alternative branches giving rise to microvillar and sustentacular lineages (Fig. 3i).

Heatmaps of dynamic gene expression revealed lineage-specific transcriptional programs along inferred trajectories (Fig. 3j). The neuronal lineage showed a transition from progenitor-associated transcription factors to neuronal maturation genes, whereas the microvillar lineage displayed early activation of epithelial differentiation and membrane trafficking, and sustentacular cells progressively upregulated support-cell programs linked to detoxification, structural integrity, and signaling (Fig. 3j).

**Fig. 2 | Composition of the human olfactory system at PCW7-PCW12.**
**a** Schematic of fetal olfactory and respiratory epithelia and stroma reconstructed from snRNA-seq, with cell types annotated by marker genes. Figure created in BioRender, and is licensed under CC BY 4.0 (https://BioRender.com/9mwoywy). **b** UMAP of the main olfactory epithelium, showing olfactory horizontal basal cells (OHBC), cycling basal cells (CyclBasal), microvillar cells (MV), sustentacular cells (SUS), globose basal cells (GBC), immediate neuronal precursors (INP), and immature olfactory sensory neurons (iOSN). **c** Dot plot of markers distinguishing RHBC, OHBC, and GBC; dot size shows expressing-cell fraction, color shows scaled expression. **d** Violin plot of per-cell *MEG3* expression in OHBC ($n = 549$ cells) and RHBC ($n = 2,588$ cells; $n = 8$ fetuses). Values were log-normalized in Seurat. Statistical significance was assessed by a two-sided Wilcoxon rank-sum test ($****P = 8.46 \times 10^{-9}$). **e** Pseudobulk cell-type proportions per specimen across developmental stages. Bars represent the mean proportion per stage, error bars indicate SEM, and points show individual specimen. **f** Marker validation by immunofluorescence in coronal sections of olfactory and respiratory epithelium at PCW9. OLIG2 labels iOSNs. Nuclei are stained with DAPI (blue). **g** Multiplex in situ hybridization at PCW9 validating markers: *MUCSAC* labels RE and *GNRH1* labels migratory GnRH1 cells. Dashed lines mark RE/OE and VNO boundaries. Nuclei: DAPI. **h** Transmission electron microscopy of the OE at PCW7.5 reveals early epithelial specialization, identifying MV, SUS, OSNs, and basal cells. The right panel shows a 90°-rotated high-magnification view of the boxed area with pseudocolors indicating cell types. **i** Immunostaining at PCW7.5 shows TUJ1+ axons emerging from the OE toward the olfactory bulb. Box 1 highlights TUJ1+ neuronal layers and SOX2+ basal/intermediate progenitors near the olfactory lumen; Box 2 shows RE with abundant KRT5+ basal cells. **j** Representative PCW12 coronal nasal section immunostained for KRT5, SOX2, and DAPI, with high-magnification views of regions 1 and 2. OE olfactory epithelium, RE respiratory epithelium, NS nasal septum, VNO vomeronasal organ, ON olfactory nerve, OL olfactory lumen. Source are provided as Source Data file.

## Transcriptional regulators and gene networks shaping human OE development

To define regulatory mechanisms underlying cell fate specification in the developing human OE, we inferred transcription factor (TF) activity and reconstructed gene regulatory networks across cell states. Comparison of TF regulon activity revealed distinct, lineage-specific regulatory programs (Fig. 4a; Supplementary Data 7). Quiescent OHBCs showed enrichment of *KLF15*, *ZIC2*, and *EGR2*, whereas GBC and cycling basal cells were marked by proliferation- and progenitor-associated TFs, including *E2F1/3/4*, *SOX2*, *ID1*, and *ID3*. In contrast, OSN displayed activation of neurogenic TFs such as *ASCL1*, *NEUROD1*, *NEUROG1*, *SOX4*, and *ATOH1*, consistent with neuronal differentiation. These regulatory programs also exhibited developmental stage specificity, with early progenitors (PCW7-8) differing from more differentiated cells at later stages (PCW10-12).

To characterize the regulatory landscape, we computed regulon activities across all cell types (Fig. 4b). Neuronal populations showed elevated activity and expression of neurodevelopmental regulators, including *PHOX2A*, *OTX2*, *NEUROD2*, *ARID1B*, *NFKBIZ*, *BARX2*, and *HOXC8*, which were largely inactive in basal and supporting cells (Fig. 4b). This combination suggests that INPs and iOSNs are governed by a multilayered transcriptional network that integrates neuronal differentiation, epigenetic remodeling, and environmental responsiveness.

OR expression in OSN depends on long-range interactions with enhancer elements known as Greek islands (GIs)[11], which in rodents are assembled into a regulatory hub by EBF1, LHX2, and the coactivator LDB1[11,12]. Single-cell studies show that OSN progenitors transiently co-express multiple ORs before resolving to monogenic expression in mature neurons[8]. Consistent with this model, we observed enriched *EBF1* activity in INPs iOSNs in the developing human OE, together with *LHX2* (Fig. 4a, b).

To integrate these findings, we reconstructed a TF-centered regulatory network for the OE (Fig. 4c). The network revealed density interconnected modules organized around key neurogenic TFs, which predominantly regulated genes associated with neuronal differentiation, axon guidance, and sensory function, whereas progenitor- and support-associated TFs regulated genes linked to cell cycle control, epithelial organization, and immune-related pathways.

## Spatial transcriptomics uncovers regionalized OE organization

We next used spatial transcriptomics to resolve the spatial organization and tissue architecture of the human fetal olfactory epithelium. To contextualize cellular populations within their native microenvironment, we applied MERFISH to coronal sections of human fetal heads, analyzing anterior and posterior regions of the nasal epithelium from $N = 1$ PCW9 fetus and a posterior section from $N = 1$ PCW11 fetus (Fig. 5a). MERFISH enables high-resolution, single-cell level in situ detection and quantification of hundreds to thousands of transcripts[19,37]. We designed a custom panel of 294 genes, comprising cell-type markers identified by snRNA-seq and 57 OR transcripts selected based on their expression profiles.

This analysis generated a dataset of 489,030 cells comprising 58,231,407 detected transcripts. Restricting the analysis to anatomically defined olfactory epithelium regions yielded 188,803 cells and 16,825,698 transcripts (mean 89 transcripts per cell) (Fig. 5b, c). To enable integrated visualization across samples, we applied Harmony for batch correction and dimensionality reduction, producing a unified UMAP representation of the spatial transcriptomic landscape (Fig. 5d).

Subsequently, we performed automated cell type annotation by transferring labels from our snRNA-seq dataset to the MERFISH data using the scMusketeer method[38], which leverages the shared 294-gene expression profile between the two modalities. This label transfer approach proved to be robust and the resulting spatial cell type assignments accurately recapitulated the known layered architecture and zonation of the olfactory epithelium (Fig. 5e).

Key signaling molecules, including PAX7, PAX6, LHX2, and members of the BMP family, play central roles in olfactory system patterning, morphogenesis, and neuronal differentiation, as well as in guiding OSN axonal projections to the olfactory bulb [16,18,33,39-41]. Using MERFISH, we observed that in human fetal tissue, consistent with findings in rodents[16,41,42], *PAX7* and *BMP4* demarcate distinct mesenchymal domains, *PAX6* is expressed in both olfactory and respiratory epithelia, and *LHX2* is selectively enriched in iOSN (Fig. 5f). These spatial expression patterns mirror those reported in mouse embryos[16,39,40,43-45], supporting conservation of key patterning programs across species.

Additional markers, including *ELF3* (respiratory epithelial duct cells), *FOXO1* (olfactory progenitors and basal cells), and *LHX2* (iOSN), further delineate the boundary between the olfactory and respiratory epithelia (Fig. 5g). This distinction was reinforced by the spatial localization of OR genes, which were restricted to the olfactory epithelium (Fig. 5h).

To further resolve OE spatial organization, we compared two anatomically distinct regions, an anterior and a posterior OE "spine", defined as narrow, radially organized domains extending from the basal lamina to the lumen and distributed along the dorso-ventral axis (Fig. 5i-k; Supplementary Fig. 8a). Focusing on these spatially coherent units enabled assessment of local tissue architecture and cell-type layering across anterior-posterior and ventro-dorsal positions.

Consistent with previous human studies[25,26], we identified the vomeronasal organ (VNO) and the associated MM at PCW9, predominantly in the most anterior sections (Fig. 5i). In both anterior and posterior regions, a clear ventro-dorsal transition RE to OE was evident (Fig. 5j, k). The RE formed a lateral domain enriched in multiciliated, deuterosomal, and basal KRT+ cells, whereas the OE was organized into olfactory-specific layers comprising OHBC, INP, iOSN and SUS (Fig. 5j, k).

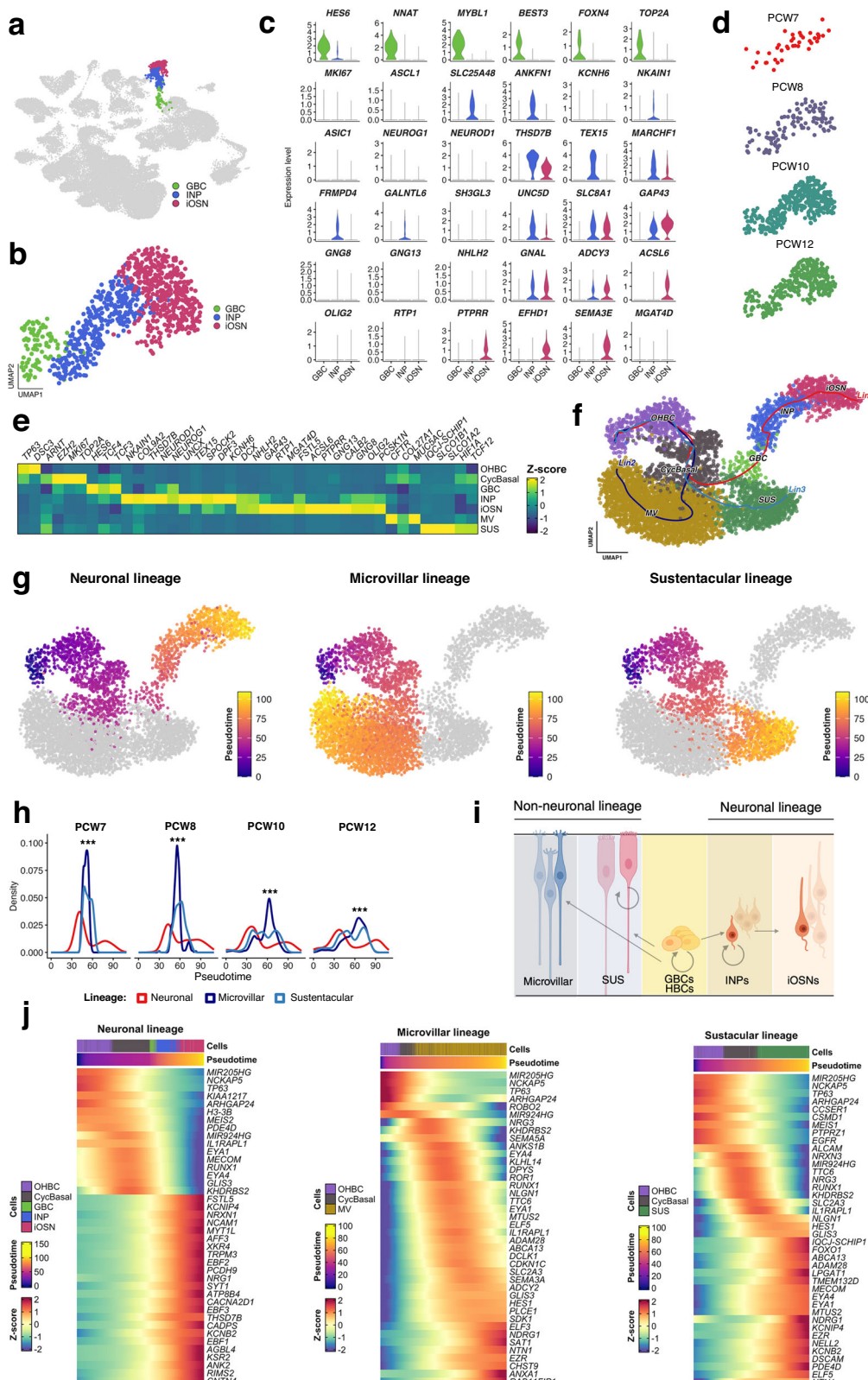

Although anterior and posterior OE regions followed similar developmental programs, they displayed marked regional differences in epithelial architecture. The anterior OE was more compact, with OSNs densely concentrated at the dorsal tip and a sharply defined boundary separating OE from RE (Fig. 5j). In contrast, the posterior OE exhibited a broader olfactory domain with a more gradual RE-OE transition and an expanded distribution of OSN and SUS (Fig. 5k). Across both regions, INP and OSN were enriched dorsally, suggesting that OE expansion

proceeds along a ventral-to-dorsal axis, elongating the epithelium, and a medial-to-lateral axis, increasing epithelial thickness.

## Olfactory receptor gene expression patterns during human OE development

The transcriptomic analysis of the fetal snRNAseq dataset revealed the expression of 169 OR genes in INP and iOSN at 4 stages of prenatal development (Fig. 6a; Supplementary Data 8). In addition, we detected

**Fig. 3 | Lineage inference and developmental dynamics of the human fetal olfactory epithelium. a** UMAP visualization of olfactory epithelial cell populations highlighting globose basal cells (GBC), immediate neuronal precursors (INP), and immature olfactory sensory neurons (iOSN). **b** UMAP subset restricted to the neuronal lineage comprising GBC, INP, and iOSN populations. **c** Violin plots showing log-normalized expression of lineage-specific marker genes in GBC, INP, and iOSN populations. Violins represent the distribution of expression values across cells. **d** UMAP projection of olfactory epithelial cells colored by post-conceptional week (PCW7-PCW12). **e** Heatmap showing the expression of selected marker genes across olfactory epithelial cell populations. **f** Slingshot-inferred developmental trajectories of the human fetal olfactory epithelium. Trajectories were computed in principal component space and projected onto the UMAP embedding for visualization. Three major lineage branches originate from olfactory horizontal basal cells (OHBC): neuronal (OHBC → iOSN), sustentacular (OHBC → SUS), and microvillar (OHBC → MV) lineages. Lineage paths were smoothed along pseudotime using generalized additive models (GAMs), and individual cells are shown as points colored by annotated cell type. **g** UMAP embedding with Slingshot-inferred pseudotime trajectories overlaid. Pseudotime progresses from early (blue) to late (yellow). **h** Density plots showing the distribution of Slingshot-inferred pseudotime values for neuronal, microvillar, and sustentacular lineages across developmental stages (PCW7-PCW12). Curves are colored by lineage and faceted by PCW. Statistical significance was assessed using Kruskal-Walli's tests (***$FDR <$ 0.001). **i** Schematic overview of neuronal (GBC → iOSN) and non-neuronal (GBC → MV) differentiation trajectories. Figure created in BioRender and licensed under CC BY 4.0 (https://BioRender.com/7m9re3j). **j** Heatmap showing temporal gene expression dynamics along pseudotime for neuronal, microvillar, and sustentacular lineages. Temporally regulated genes were identified by fitting generalized additive models to test for smooth expression changes along pseudotime, followed by filtering for expression level, pseudotime coverage, and dynamic range (log fold change ≥0.5). Genes with FDR-adjusted $q \leq 0.05$ were considered significant. Expression values are log-normalized and z-scaled per gene for visualization. Source data are provided as a Source Data file.

the expression of the vomeronasal type-1 receptor 1 (*VN1R1*) in a single OSN (Fig. 6a).

OR transcripts were virtually absent from non-neuronal populations. In contrast, OR expression was strongly enriched in iOSN, with substantially lower levels detected in INP (Fig. 6b). We then assessed the proportion of INP and iOSN expressing the ORs (Fig. 6c). INPs predominantly lacked OR expression, whereas iOSN displayed a significant shift toward OR-positive states. We observed that over 40% of OR-positive iOSN expressed only a single receptor, with lower proportions of dual expression and rare cases of cells expressing three or more ORs (Fig. 6c; Supplementary Fig. 9a; Supplementary Data 8). These data indicate that OR expression initiates during the transition from precursor to immature neuron and rapidly converges toward restricted receptor usage, with only 10% of iOSN expressing 2 ORs and <1% expressing 3 or more ORs (Fig. 6c), consistent with the progressive establishment of the "one-neuron/one-receptor" rule[4,6] during prenatal human development.

Both class I and class II ORs were identified, with a predominance of Class II ORs (Fig. 6d; Supplementary Data 9), mirroring the situation of the adult human OE[29].

Additionally, we detected OR transcripts in some non-olfactory cell types. However, OR expression was more prevalent in iOSNs compared to other cell populations within the OE and in the surround nasal territories (Fig. 6e).

We next analyzed the distribution and expression of ORs based on their genomic locations (Fig. 6f, g). The number and expressions of ORs were not evenly distributed across the genome; instead, OR loci are denser in chromosomes 1, 7, 11, 12, and 19 (Fig. 6f). However, these loci are not necessarily more transcriptionally active (Fig. 6g) suggesting that some OR gene choice is already active locally in each genomic loci, in order to drive the expression of a particular gene.

To assess the degree of OR dominance within individual cells, we computed OR dominance scores based on the relative expression of the most highly expressed OR compared with the second-ranked OR (Fig. 6h; Supplementary Data 10). Cells with low dominance scores were primarily non-neuronal cells and INP, whereas higher dominance scores were almost exclusively associated with iOSN. The proportion of iOSN increased progressively across higher dominance score bins, indicating strengthening dominance of a single OR as neuronal maturation proceeds. Consistent with this, the number of high-dominance cells increased markedly over developmental time (Fig. 6i). High-dominance cells were virtually absent at PCW7, increased at PCW8, and became abundant by PCW10 and PCW12, predominantly within the iOSN population.

We then identified the OR genes most frequently detected as dominant across high-dominance cells (Fig. 6j; Supplementary Fig. 9b). Few dominant OR genes were detected in iOSN at PCW7 and PCW8,

consistent with the low overall frequency of OR expression at these early time points (Supplementary Fig. 9b). By PCW10, however, a clear set of dominant OR genes emerged, each represented by multiple high-dominance cells. This repertoire expanded further by PCW12, with an increased number of OR genes showing repeated high-dominance expression across cells.

Finally, to investigate the pattern of OR expression during human fetal development, we analyzed OR transcription across individual cells (INP and iOSN). Ordering cells by their dominant OR revealed a striking diagonal pattern of OR expression, indicating that individual neurons preferentially express a single OR gene (Supplementary Fig. 10). Notably, INP cells exhibited limited OR expression, whereas robust expression was confined to iOSN. The frequency and strength of dominant OR expression increased from PCW7 to PCW12, indicating progressive stabilization of OR choice over developmental time. Together, these data demonstrate that the hallmark feature of OSN, singular OR expression, is established early in human fetal development and becomes increasingly refined as neurons differentiate.

### Spatially resolved olfactory epithelium OR genes profiling confirms the 1 OR-1 OSN rule at single-cell resolution

To gain spatial insight into OR gene expression during human fetal development, we applied MERFISH to map OR transcripts within the OE at high resolution. MERFISH further delineated OR expression in the OE of two human fetal samples at PCW9 and PCW11 (Supplementary Fig. 11a–c).

We first examined the spatial organization of the OE by integrating cell-cell transcriptomic correlation with physical proximity to define discrete spatial domains (Fig. 7a). This analysis revealed a highly specific domain, D1016, corresponding to the OE and RE. Domain D1016 was surrounded by domain D1013, a more heterogeneous region that includes all OE cell types along with OECs and Schwann cells, suggestive of a supportive or transitional niche. Additional spatial domains were clearly delineated and corresponded to known anatomical features: domain D1015 encompassed cartilage-associated cells, D1014 was enriched in osteoblasts, and D1010 corresponded to stromal compartments (Fig. 7b).

To assess adherence to the 1 OR-1 OSN rule, we developed an adjusted dominance score, defined as the expression difference between the most and second most highly expressed OR gene within each cell, scaled by the logarithm of the expression level of the top-expressed OR. This metric accounts for both dominance and expression magnitude, thus minimizing noise from lowly expressed transcripts. Application of this score across the dataset revealed a distinct population of high-scoring cells localized primarily within the OE (Fig. 7c). Cell types with a high dominance score mostly corresponded to iOSN and it increased across developmental stages, reaching a

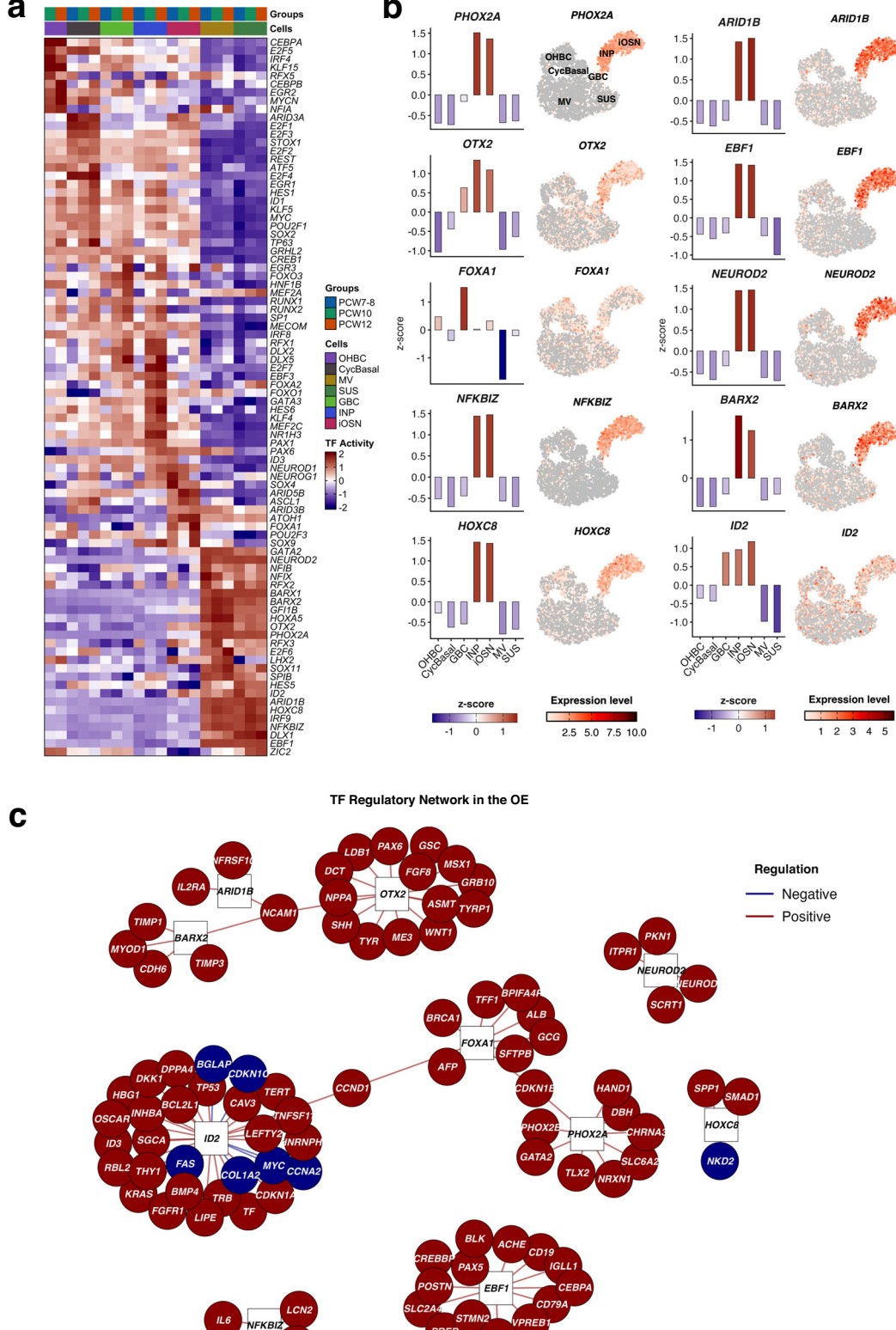

maximum at PCW11 (Fig. 7d), thus providing an additional demonstration of the single-OR expression rule in OSNs.

Hierarchical clustering of OR expression across spatial spots revealed coherent patterns of OR gene usage associated with specific cell types (Fig. 7e). iOSN exhibited enriched expression of multiple OR genes relative to other OE populations, while non-neuronal cells showed little to no OR signal.

We visualized individual OR transcripts alongside canonical OSN markers using MERFISH at PCW9 and PCW11 (Fig. 7f). OR genes were detected in spatially discrete neuronal clusters, co-localizing with the OSN markers, including *RTN1*, consistent with spatial segregation and early onset of OR gene choice.

At PCW9, OR expression was sparse, with the great majority of OSNs expressing 1 single OR and some OSNs co-expressing two

**Fig. 4 | Transcription factors (TFs) activities in olfactory sensory epithelium.** **a** Heatmap of transcription factor (TF) regulon activities across developmental stages in the human olfactory epithelium (PCW7-12). TF activities were inferred from single-nucleus RNA data using the tfsulm assay, which quantifies each TF's regulatory influence on its predicted target genes based on a curated regulatory network. Activity scores were row-wise Z-score normalized across cells, highlighting relative activity dynamics for each TF. Rows correspond to TFs, columns correspond to cell type-stage combinations, and color intensity indicates scaled TF activity. **b** Bar plots showing TF activity (left) and corresponding gene expression (right) across olfactory cell types. TF activity was inferred from the tfsulm assay, while gene expression was measured from RNA assays. Each row represents one TF; color intensity indicates scaled activity or expression. Distinct activity and expression patterns reveal lineage-specific regulatory programs in neuronal and supporting cells. **c** Directed regulatory network of TF interactions in the developing olfactory system. Source TFs are represented as squares, with top predicted target genes shown as circles. Edges denote regulatory interactions, colored by mode of regulation (red = activation, blue = repression) and scaled in transparency by absolute regulatory score (|mor|). Node colors indicate regulatory type, and text color differentiates sources (black) from targets (white). Source data are provided as a Source Data File.

distinct OR genes, consistent with the dominance and co-expression patterns inferred from transcriptomic data.

In PCW11 samples, OR expression became more prevalent, with a greater number and diversity of OR genes detected across the OE (Fig. 7f). Strikingly, the co-expression of multiple ORs within single OSNs was rarely observed at this later stage, suggesting a developmental refinement of OR gene regulation.

Together, these findings demonstrate that OR expression in the human fetal olfactory epithelium is spatially patterned, cell-type restricted, and progressively stabilized into dominant expression states within defined epithelial domains, supporting an early emergence of organized receptor expression during human olfactory system development.

## Discussion

The nasal region comprises cartilage, bone, mesenchyme, neural crest-derived glia, and placode-derived sensory and endocrine neurons, yet its development and molecular organization remain poorly understood, largely due to limited access to fetal tissue and its complex spatial architecture.

In this study, we present a combined single-nucleus and spatial transcriptomic atlas of the human fetal nasal region spanning PCW7-12. This work provides a systematic view of the developmental dynamics of the diverse cell populations that compose the human nasal region during organogenesis. Our integrated analyses reveal marked temporal shifts in cellular composition, with progenitor populations predominating at early stages (PCW7-9), followed by progressive expansion of mesenchymal and osteogenic subtypes, consistent with active craniofacial growth and tissue remodeling and establishing a resource for future studies of human nasal development.

A key feature of the OE, that has been well documented in rodents, primates, and humans, is its capacity for lifelong neurogenesis, maintaining a consistent population of OSNs throughout life[8,29,34,35,46–48]. This renewal is driven by GBCs, which function as actively proliferating progenitors[34,49,50]. In rodents, following injury, the OE demonstrates regenerative ability, with normally quiescent OHBCs becoming activated to differentiate and replenish all major epithelial cell types[35,51].

Trajectory inference and pseudotime analyses performed in this study highlight the bipotent nature of OHBCs and GBCs, which are predicted to give rise to both neuronal and non-neuronal lineages, including sustentacular and microvillar cells. These findings suggest that the regenerative architecture of the OE is established early during human development.

Consistent with this, our data reveal notable differences in the timing and cellular complexity of human OE development compared to rodents. In mice, early and mid-embryonic stages of OE development are characterized by limited cellular diversity, with full stratification and emergence of all major cell types occurring late in gestation[18]. By contrast, in humans, nearly all major OE cell types present in the adult, except Bowman's glands, are already detectable during early fetal stages. Whether the absence of Bowman's gland cells reflects technical limitations or later developmental onset remains to be determined. Notably, whereas murine OHBC are thought to emerge only shortly before birth[17,18], we demonstrate that OHBC are present during early human OE development. This early emergence points to species-specific differences in basal cell deployment and suggests that HBC may play broader developmental roles in humans beyond injury-induced regeneration.

Our study also identifies molecular features associated with early olfactory neuron maturation. We report *PTPRR* as a robust marker of iOSN. PTPRR is a negative regulator of MAPK/ERK signaling and has been implicated in neuronal differentiation and synaptic plasticity in the central nervous system[52,53]. Its expression in fetal human iOSN, together with its presence in adult pig and human OE datasets[29,54], suggests a conserved role in OSN maturation or signaling modulation. In parallel, expression of *GAP43* and *SEMA3E* is consistent with ongoing neurite outgrowth and axonal guidance, while expression of *GNAL*, *ADCY3*, *LHX2*, and *EBF1* indicates early activation of the canonical olfactory transduction pathway prior to full neuronal maturation.

Spatial transcriptomic analyses further revealed pronounced regional organization within the developing OE. We observed clear anterior-posterior and ventro-dorsal differences in epithelial layering and cellular composition. Anterior regions exhibited compact olfactory domains with sharply defined boundaries, whereas posterior regions showed expanded sensory territories and more gradual transitions between olfactory and respiratory epithelia. These spatial gradients likely reflect region-specific inductive environments, supported by the patterned expression of key developmental regulators[16] such as *PAX6*, *PAX7*, *BMP4*, and *LHX2*.

Importantly, spatial mapping also provided insight into the early establishment of OR expression patterns. Using MERFISH, we show that ORs are expressed in spatially restricted epithelial domains, suggesting that elements of the OR map are established prenatally. However, because these analyses rely only on two donor specimens and a limited set of OR genes, these conclusions should be interpreted cautiously and validated in larger cohorts.

In rodents, OR gene choice is governed by complex regulatory mechanisms, including enhancer elements and multi-chromosomal enhancer hubs that enforce singular OR expression per neuron[9–12]. Our integration of single-nucleus and spatial transcriptomics data demonstrates that analogous organizational principles probably operate in humans.

By combining single-nucleus transcriptomics with high-resolution spatial mapping, our study reveals that key principles of OR regulation are established early during human prenatal development and are tightly linked to both neuronal differentiation and tissue architecture. Transcriptomic analyses identified expression of 169 OR genes, compared to the 545 reported in the adult OE[29], with OR activation largely restricted to iOSN and emerging during the transition from INPs. We also detected the expression of *VN1R1*, equally reported in a dataset of adult human OE[29]. Although we detected *VN1R1* in a single OSN, we cannot exclude broader expression of vomeronasal receptors or trace amino acid receptors, as these may occur in rare cell populations that were under-represented or lost during tissue sampling and dissociation.

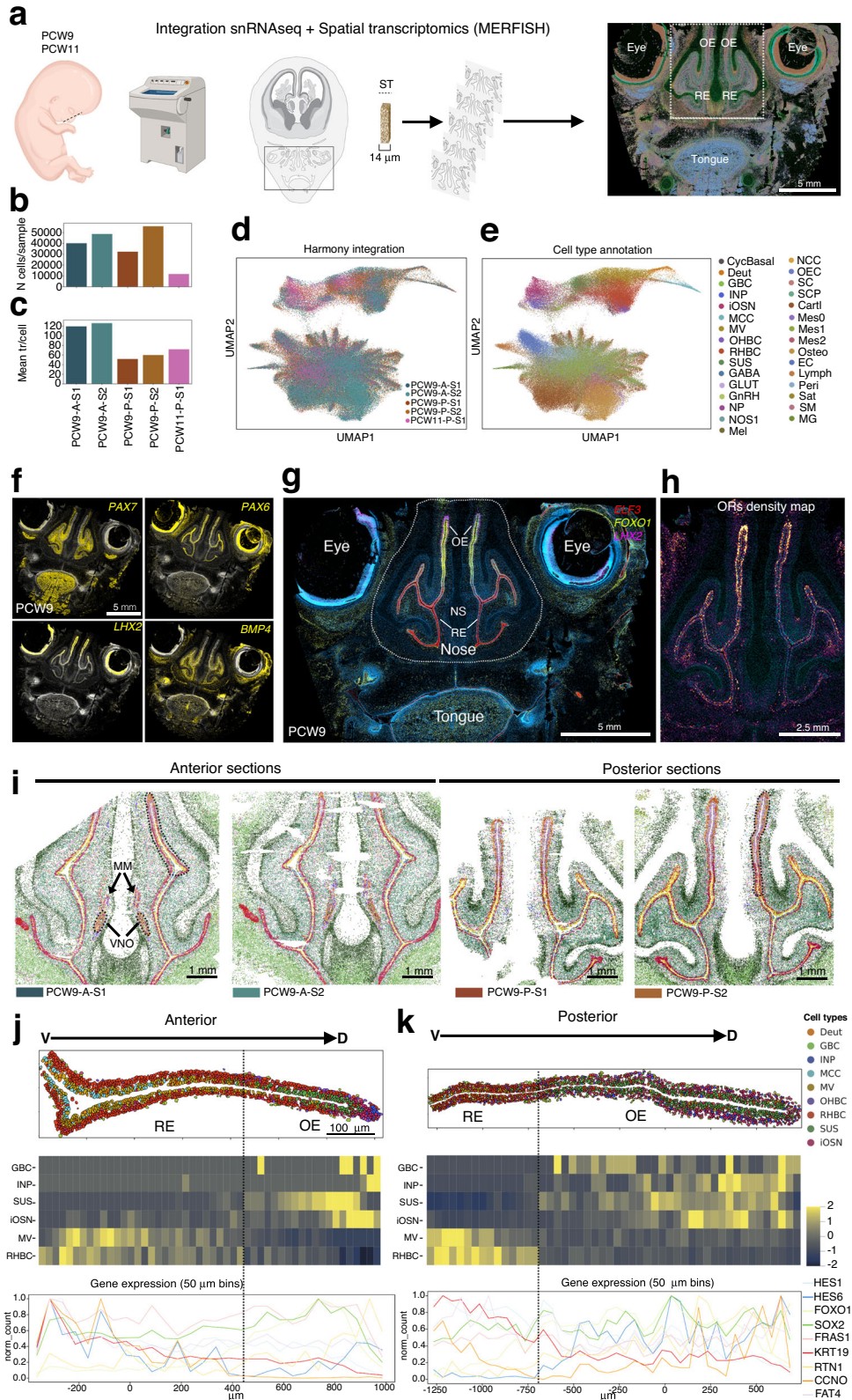

At the single-cell level, OR expression rapidly converged toward restricted receptor usage, with most OR-positive iOSN expressing a single receptor and only rare instances of multi-OR co-expression, consistent with the progressive establishment of the one-neuron-one-receptor rule well before birth. This developmental refinement was further supported by increasing OR dominance scores over time and the emergence of a reproducible subset of dominant OR genes by PCW10-12. The predominance of class II ORs mirrors the adult human olfactory epithelium[29], suggesting early specification of receptor class usage.

Spatial transcriptomic analyses extended these findings by demonstrating that OR expression is not only cell-type specific but also spatially organized within the OE. OR transcripts localized to discrete epithelial domains corresponding to the OE and were largely excluded

**Fig. 5 | A spatially resolved single-cell atlas of human fetal olfactory development. a** Schematic of sampling and workflow of spatial transcriptomics. Figure created in BioRender and licensed under CC BY 4.0 (https://BioRender.com/c1vk104). **b** Total number of spatial cells per sample. **c** Mean number of detected transcripts per cell per sample. **d** Harmony integration UMAP colored by specimen. **e** Cell type annotation UMAP colored by cell type obtained using automatic label transfer method using snRNA-seq as the reference. **f** Image of a coronal head section at PCW9, covering the nasal region, imaged with MERFISH using *PAX7*, *PAX6*, *LHX2* and *BMP4* probes. **g** Same section as in **f** imaged with MERFISH using different probes (*ELF3*, *FOXO1*, *LHX2*) to spatially define the olfactory and respiratory epithelia. DAPI nuclear staining was used as counterstaining. **h** Higher magnification of the region depicted in **g** within the dotted line showing the olfactory receptor

density map in the olfactory epithelium (OE). **i** Spatial visualization of the 4 sections (2 for anterior and 2 for posterior sections at PCW9). **j, k** Spatial distribution of cell types and gene expression along the anterior-posterior axis of the human fetal olfactory epithelium (dotted line indicated in panel **i**) of a PCW9 sample. The dotted line marks the boundary between the respiratory epithelium (RE) and the olfactory epithelium (OE). Top panels show cell type organization. Middle panels: heatmaps showing the evolution per cell type along the olfactory epithelium region. The signal is normalized by cell type (row-scaled). GBC, INP, SUS, and iOSN correspond to OE cell types, whereas MV and RHBC correspond to RE cell types. For each heatmap, the region was divided into 50 equally sized bins. Bottom panels indicate the evolution of the gene expression for a panel of gene in our ST panel.

from surrounding non-olfactory regions, reinforcing the notion that OR gene choice is tightly coupled to epithelial identity. High OR dominance scores were spatially confined to iOSN within the OE and increased between PCW9 and PCW11, providing independent spatial validation of the single-OR expression rule.

Together, these results indicate that human OR gene choice is an early, developmentally regulated process that unfolds within defined spatial and cellular contexts. The concordance between transcriptomic and spatial data supports a model in which OR activation, dominance, and stabilization emerge in parallel with neuronal maturation and epithelial patterning, establishing core organizational features of the human olfactory system well before birth.

In summary, our study defines cellular trajectories, spatial architectures, and molecular programs that govern human olfactory neurogenesis and epithelial organization during early development. By integrating single-nucleus and spatial transcriptomics, we uncover both conserved mechanisms and human-specific features of olfactory system formation. This atlas provides a framework for investigating congenital anosmia, craniofacial disorders, and the developmental origins of sensory and respiratory epithelial specialization in the human nose.

## Methods

### Ethical statement
The human fetal tissues used in this study were made available in accordance with French bylaws (Good Practice Concerning the Conservation, Transformation, and Transportation of Human Tissue to Be Used Therapeutically, published on December 29, 1998). Authorization to use human tissues was granted by the French agency for biomedical research (Agence de la Biomédecine, Saint-Denis La Plaine, France; N° PFS19-012) and the INSERM Ethics Committee (IRB00003888). All samples were provided by the INSERM HuDeCa Biobank and used in full compliance with national regulations.

### Human embryos and fetal samples
The human fetuses included in this study were obtained legally from voluntary abortions, with post-conceptional weeks ranging from 7 to 12 ($n = 1$ PCW7 XY, $n = 1$ PCW8 XX, $n = 1$ PCW10 XX, $n = 1$ PCW10 XY, $n = 1$ PCW10.5 XY, $n = 2$ PCW12 XY, $n = 1$ PCW12 XX), from voluntarily terminated pregnancies upon obtaining written informed consent from the donors (Gynaecology Department, Jeanne de Flandre Hospital, Lille, France). All specimens were initially selected based on macroscopic morphological criteria, excluding samples with obvious malformations. In compliance with French privacy regulations, no personal or identifying donor information (including ethnicity, race, genetic background, or date of birth) was available.

### Sample collection and single-nucleus RNA-seq
Frozen fetal nasal tissues were stored at −80 °C in isopentane cooled with liquid nitrogen. RNA quality was assessed from 70-µm cryosections following nuclei isolation and RNA extraction; only samples with RNA integrity number (RIN) ≥ 7, measured using an Agilent

Bioanalyzer, were used. snRNA-seq was performed on eight frozen human fetal nose samples. Tissues were coronally cryosectioned, and five consecutive 70-µm sections (350 µm total) were collected once the olfactory sensory epithelium became visible. Regions containing olfactory and respiratory epithelia and surrounding mesenchyme were microdissected, snap-frozen, and stored at −80 °C. Samples were multiplexed into two pools of four samples each, and individual identities were recovered using single-nucleotide polymorphisms (SNP). Frozen sections from four samples were homogenized in 800 µL NP40 lysis buffer (10 mM Tris-HCl pH 7.4, 10 mM NaCl, 3 mM MgCl$_2$, 0.1% NP40, 1 mM DTT) supplemented with RNase inhibitors (RNaseIn®, Promega; RiboLock®, Thermo Fisher Scientific; 0.4 U/µL each) using a Dounce homogenizer (10 strokes loose pestle, 10 strokes tight pestle). Nuclei were incubated on ice (4 min), filtered (40 µm), washed, and centrifuged (500 × $g$, 5 min, 4 °C). Pellets were resuspended in wash buffer (PBS, 1.5% BSA, RNase inhibitors), filtered (5 µm), centrifuged, and resuspended in 30 µL. Nuclei were counted using a Countess II FL and loaded onto a Chromium Controller (10x Genomics). Libraries were prepared using Chromium Next GEM Single Cell 3′ v3.1 chemistry and sequenced on a NextSeq 2000 Illumina platform. Raw sequences Illumina were stored in BCL (Base Call) files, which contain raw fluorescence intensity signals and base call information for each sequencing cycle. BCL files were converted to FASTQ using Cell Ranger *mkfastq*[55] and processed with Cell Ranger v6.0.0. Intronic reads were retained, and reads were aligned to the hg38 reference genome using the STAR[56]. Gene expression was quantified using unique molecular identifiers, low-quality nuclei were filtered, and gene–barcode matrices were generated.

### Bulk RNA-seq QC and reference for demultiplexing
Bulk RNA-seq data were processed using bcftools (https://doi.org/10.1093/gigascience/giab008). Individual VCFs were indexed, merged, and filtered to retain variants with minor allele count ≥1 and genotype missingness <20%. Per-variant allele frequency, minor allele frequency, and donor-level completeness were computed using a Python workflow built on *cyvcf2* function. Genotype concordance between donors and bulk RNA-seq samples was assessed to ensure high-quality variant calls for reference in multiplexed single-nucleus demultiplexing.

### Genetic demultiplexing
Multiplexed single-nucleus data were demultiplexed using Demuxafy[20] in a Singularity container, integrating outputs from Vireo[22], Souporcell[21], and scds (Single-Cell Doublet Scorer)[57]. Donor VCFs from bulk RNA-seq were used as genotype references. Allele concordance between single-nucleus genotypes and bulk variants was computed, and donor assignments were classified as mapped, ambiguous, or rejected based on posterior donor genotype probability separation (delta genotype probability), as implemented in Vireo[22], using thresholds of minimum concordance ≥0.60 and probability delta ≤0.05. Doublets and unassigned nuclei were removed, retaining high-confidence singlets for downstream analysis. Consensus donor

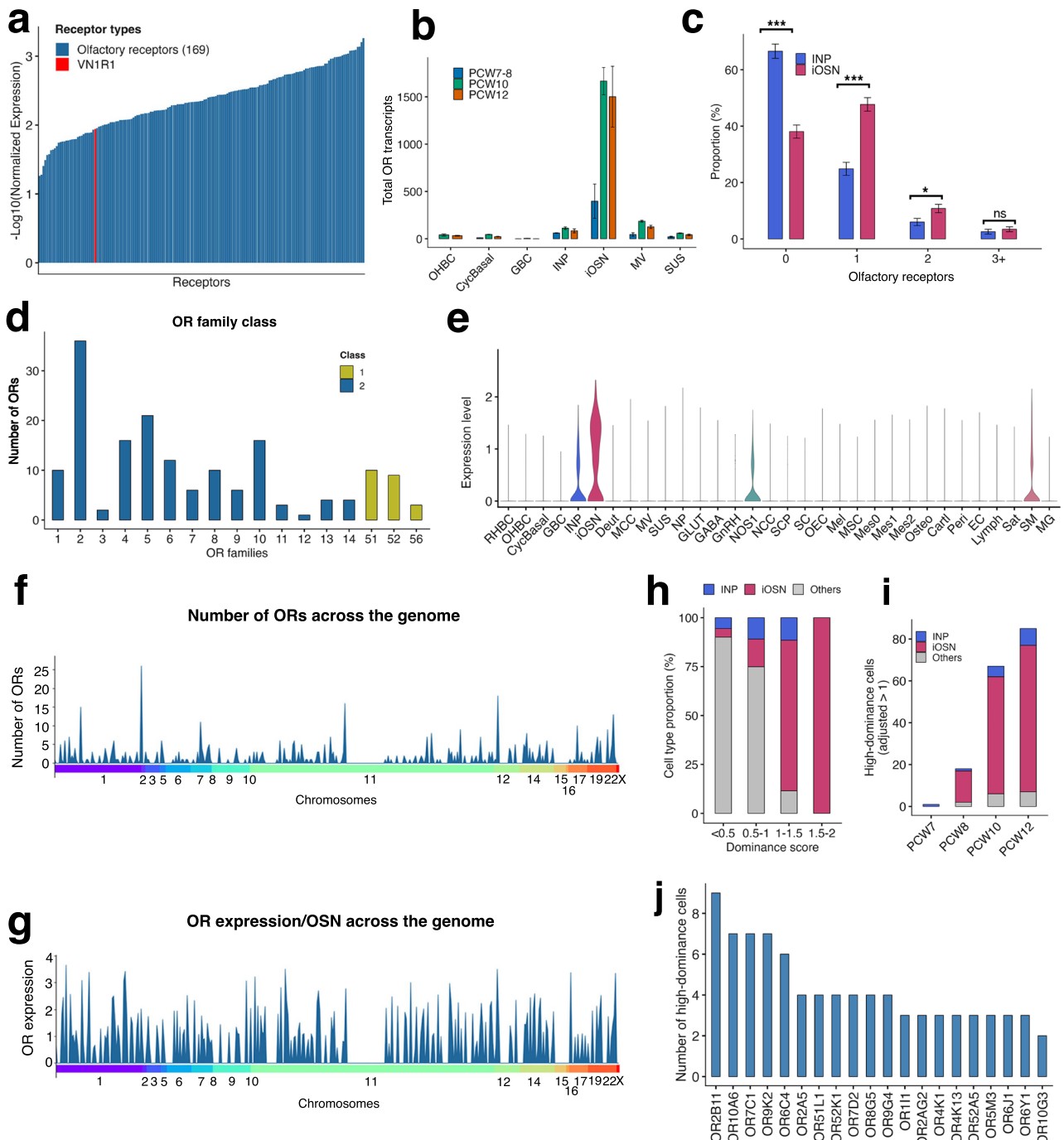

**Fig. 6 | Analysis of the expression of the ORs in the fetal human olfactory system. a** Expression levels of detected ORs ($n = 169$) in intermediate neuronal progenitors (INPs) and immature olfactory sensory neurons (iOSNs) across PCW7-12 ($n = 8$ donors). Receptors are shown on the x-axis; the y-axis indicates log-normalized expression. **b** Bar plot illustrating total olfactory receptor (OR) transcript counts across the cell types that make up the olfactory epithelium (OE) at developmental stages PCW7-12 ($n = 8$ donors), with values aggregated per donor. Values represent mean ± SEM. **c** Distribution of OR expression in INP (0 OR, $n = 233$; 1 OR, $n = 87$; 2 ORs, $n = 21$; ≥ 3 ORs, $n = 9$) and iOSN (0 OR, $n = 166$; 1 OR, $n = 208$; 2 ORs, $n = 47$; ≥ 3 ORs, $n = 15$) across the specimen ($n = 8$). Values represent mean ± SEM. A two-sided Pearson's $\chi^2$ test (no Yates correction) revealed a significant association between cell identity and OR distribution (Cramér's $V = 0.285$, 95% CI 0.210–0.285). *Post hoc* per-bin $\chi^2$ tests were corrected using the Benjamini-Hochberg method (0 OR: \*\*\*padj = 7.92e−15; 1 OR: padj = 9.72e−1; 2 ORs: \*padj = 0.0238; ≥ 3 ORs: padj = 0.482; ns: not significant). **d** Number of OR by family in iOSN and INP. **e** Violin plot showing the distribution of ORs in olfactory and non-olfactory cells from the fetal nasal region. **f** Genomic distribution of OR genes across chromosomes. **g** Chromosomal mapping of OR expression levels per OSN. **h** Stacked bar plot showing the proportion of cells per OR dominance score bin, stratified by OE cell identity. Dominance scores were computed based on the ratio between the top-expressed and second-highest OR per cell, adjusted for sparse single-cell expression. Bins were defined using fixed thresholds (≤0.5, 0.5–1, 1–1.5, 1.5–2, >2). (**i** Number of high-dominance cells (dominance score > 1, top OR ≥ 1, and ≤3 co-expressed ORs) per developmental stage, stratified by identity. (**j**) Top 20 OR genes most frequently observed as dominant in high-dominance cells. Bars indicate the number of cells in which each OR was the highest-expressed receptor. Source data are provided as a Source Data File.

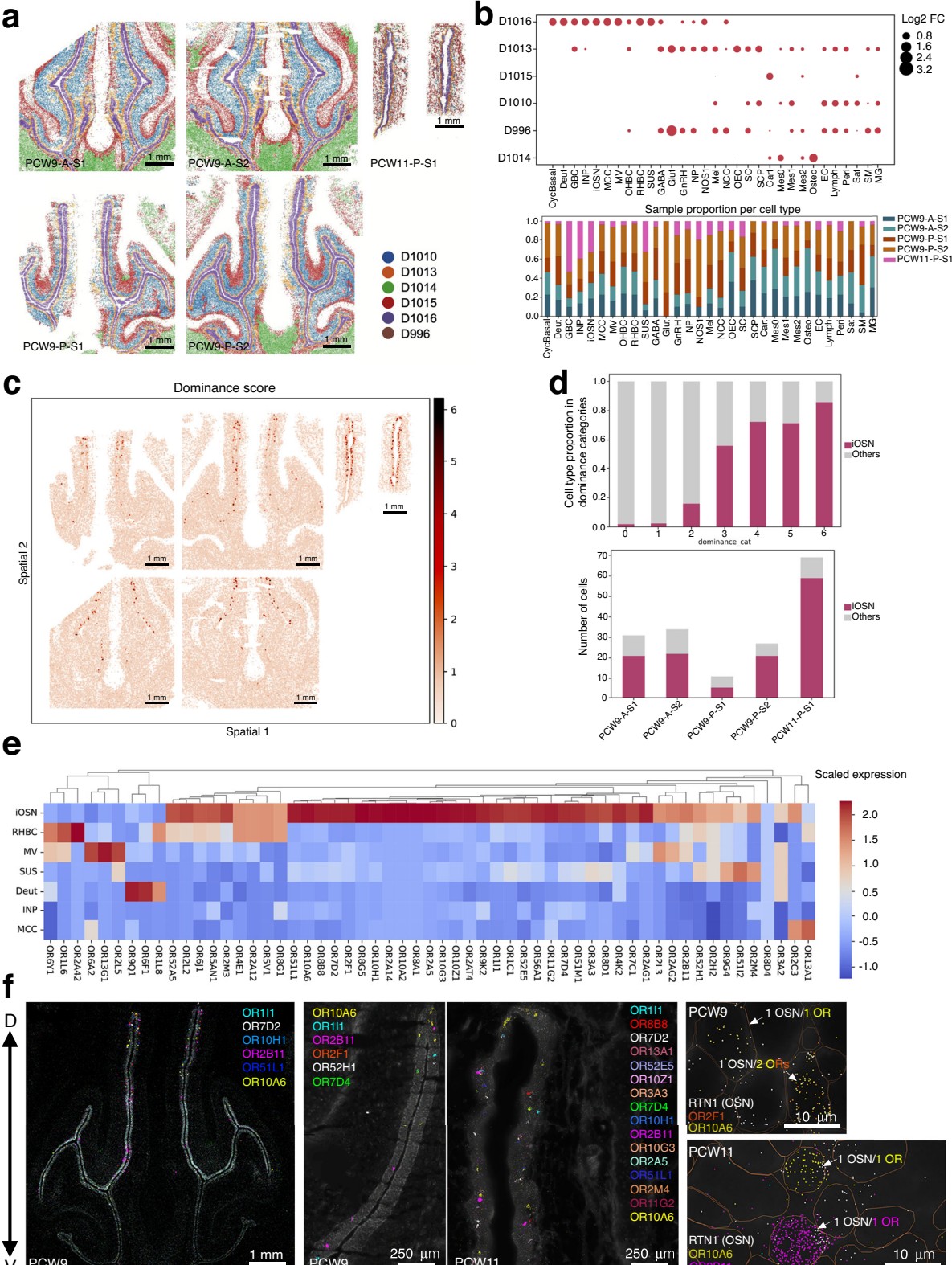

**Fig. 7 | The spatial distribution of sensory neurons and olfactory receptors in the MOE. a** Novae Spatial domains identification across sections. **b** Upper panel shows the cell type enrichment within Novae spatial domains. Results show D1016 as olfactory epithelium (OE) and D1013 surrounding the OE. Lower panel indicates the percentage per sample within each cell types. **c** Dominance score of the highest olfactory receptor compared the 2nd highest olfactory receptor per cell. **d** Upper panel: cell type proportion in dominance categories. Lower panel: number of cells per cell type of cells with a dominance score >3 within olfactory and respiratory epithelium (novae domains D1013 + D1016). **e** Heatmap showing scaled expression of representative OR genes across epithelial cell types. **f** Representative photomicrographs of a MERFISH experiment showing the spatial visualization of the listed OR's transcripts at PCW9 and PCW11.

assignments were visualized using heatmaps of Vireo–Souporcell agreement and posterior genotype probability deltas across SNPs.

## Quality control

Seurat v5.3.158[58] and R v4.5.1 were used for preprocessing. Cells were excluded if they had <500 detected genes, <500 UMIs, or >5% mitochondrial RNA. Outliers in library complexity and mitochondrial content (>5 median absolute deviations) were removed. Ambient RNA contamination was reduced using *DecontX*[59].

## Detection and removal of maternal erythroid contamination

To identify and remove maternal blood-derived contamination, we implemented a sex- and erythroid-based quality control strategy at the single-cell level. For each cell, a sex score was computed as the $\log_2$ ratio between expression of the X-linked gene *XIST* and the summed expression of Y chromosome genes (*UTY, RPS4Y1, ZFY, DDX3Y,* and *KDM5D*), using log-normalized RNA counts. Cells were classified as *female-like, male-like,* or *ambiguous* based on predefined sex score thresholds. In parallel, an erythroid contamination score was calculated as the summed expression of canonical hemoglobin genes (*HBB, HBA1, HBA2, HBE1, HBG1, HBG2,* and *HBM*), which mark maternal erythrocyte-derived ambient RNA. Cells with erythroid scores exceeding the 95th percentile of the global distribution were considered erythroid-high. Sample-level sex identity was inferred by majority voting (>50%) of sex classifications within each sample. Cells exhibiting discordant sex assignment relative to the inferred sample sex and elevated erythroid signal were flagged as contaminated. Ambiguous sex calls were optionally excluded. Flagged cells were removed prior to downstream analyses. Summary statistics, including per-sample sex composition, erythroid burden, and removal fractions, were reported for quality control and visualization. The gene descriptions are listed in Supplementary Data 11.

## Data integration and dimensionality reduction

Filtered datasets were log-normalized using *NormalizeData()*, and 2000 highly variable features were identified using the variance-stabilizing transformation method (*FindVariableFeatures()*) implemented in Seurat (https://doi.org/10.1038/nbt.3192). Batch effects were corrected using Harmony (https://doi.org/10.1038/s41592-019-0619-0), followed by principal component analysis (PCA). Uniform Manifold Approximation and Projection (UMAP) was used for visualization [https://doi.org/10.48550/arXiv.1802.03426]. Nearest-neighbor graphs were constructed, and clusters were identified using the Louvain algorithm (https://doi.org/10.1088/1742-5468/2008/10/P10008). Prior to downstream analyses, data layers were merged using *JoinLayers()*, and matrix annotations were harmonized to ensure consistency across datasets. Prior to downstream modeling analyses, layers were merged by *JoinLeyers()* and matrix names were harmonized to ensure consistent annotation across datasets. Initial automated annotation used Seurat Azimuth[60] with a fetal reference atlas[61,62], refined by canonical gene markers. The following cell types were annotated (selected markers are listed): CycBasal (*TOP2A, MKI67);* RHBC (*TP63, KRT5, KRT19, RASSF6); OHBC (TP63, KRT5, COL17A1*); Deut (*CDC20B, CCNO*); MCC (*FOXJ1, CFAP126, STOML3, DTHD1, FRMPD2, VWA3A*); MV (*SERPINB7, GABRP, CFTR, HTR1F*); SUS (*IQCJ-SCHIP1, SLCO1A2, SLCO1B1*); GBC (*HES6, NNAT, MYBL1*); INP (*NEUROD1, NEUROG1, TEX15*); iOSN (*GNAL, GNG8, MYT1L, ACSL6, GNAL, RTN1, ADCY3, GNG8, PTPRR, RTP1,OLIG2, CBX3*); GABA (*NXPH1, GAD1,GAD2*); GLUT (*SCL24A2, SLA, NEUROD6*); NP (*SHROOM3, PAX6, SOX2*); GnRH (*GNRH1, PTPRN1, SMIM35*); NOS1 (*NOS1, EML5, CHRM2*); SCP (*TOP2A,CDCA2, BUB1); SC (COL19A1, GINS3, OLFML2A*); NCC (*CDH19,THSD4,DCC*); OEC (*SLC38A11, TMOD1, NKD1*); Mel (*HEY2, MMD2, MITF*); EC (*VWF, EMCN, FLT1*); Lymph (*PROX1, STAB2*); Sat (*PAX7, PDE1C*); SM (*MYPN, AGBL1, ASB5*); MSC (*SOX2, GSTP1*); Mes0 (*TOP2A, KIF18B*); Mes1 (*ITGA8, PAX9*);

Mes2 (*CREB5, NRG1*); Peri (*PDGFRB, TRPC6, ABCC9*); Osteo (*COL13A1, ANOS*); Cartl (*ACAN, COL27A1, COL9A2*); MG (*CD14,TMEM119*).

The gene descriptions are listed in Supplementary Data 11.

## Differential expression

Differential expression and cell type marker identification were performed using Seurat FindMarkers() by the Model-based Analysis of Single-cell Transcriptomics (MAST) framework as test use. MAST models the zero-inflated nature of single-cell RNA-sequencing data with a two-part hurdle model, separately modeling gene detection and expression conditional on detection. For each comparison, the model included cellular detection rate as a covariate to account for differences in sequencing depth. Genes were considered significant at a threshold of $\alpha = 0.05$, and multiple-testing correction was applied using the Benjamini-Hochberg procedure where appropriate. Identified markers were used to define cell type-specific transcriptional signatures and guide downstream analyses.

## Donor-level cell type fraction analysis

Donor-level fractions of each cell type were calculated as the proportion of cells assigned to that cell type within each donor. Fractions were log-transformed for visualization. Differences in cell type fractions between sexes (female versus male donors) were assessed using two-sided Wilcoxon rank-sum tests.

## Pseudobulk cell type composition

Donor-level pseudobulk cell type proportions were calculated to ensure that each donor contributed a single, independent estimate of cell type composition. For each donor ($d$) and cell type ($c$), the proportion was computed as (1):

$$P_{(d,c)} = \frac{N_{d,c}}{\sum_{c'} N_{d,c'}}$$

where:

- $d$ = donor index
- $c, c'$ = cell type indices
- N($d, c$) = number of cells of type $c$ in donor $d$

Cell types present in fewer than three donors per developmental stage were excluded from stage-wise comparisons. Differences in cell type proportions across stages were assessed using the Kruskal-Wallis test.

## Trajectory inference and pseudotime

Developmental trajectories were inferred using Slingshot (v2.6.0)[63] applied to PCA embeddings generated from the top 50 highly variable genes. Following established lineage relationships in the human olfactory epithelium[28], OHBC were specified as the root population. Terminal fates were defined as immature iOSN, MV, and SUS; Slingshot inferred the branching structure without constraints, consistently returning three trajectories.

Pseudotime values were computed along principal curves for each lineage, and lineage assignments were stored at the cell level. Trajectories were visualized in UMAP space using a pseudotime color gradient. For lineage-specific dynamics, we fitted generalized additive models (GAMs) (https://www.taylorfrancis.com/books/mono/10.1201/9781315370279/generalized-additive-models-simon-wood) with smooth terms modeling pseudotime as a function of PCW. Differential pseudotime progression across PCWs was assessed per lineage with Benjamini-Hochberg FDR correction.

## Transcription factory activity

TF activity was inferred from snRNA-seq data using the DecoupleR R package (v2.12.0)[64] with the univariate linear model, incorporating

curated TF-target regulatory interactions weighted by mode of regulation. Because TF activity is inferred from downstream target gene expression, TF mRNA abundance is not expected to directly reflect regulatory activity, which may be influenced by post-transcriptional and post-translational mechanisms. Log-normalized gene expression values from the Seurat object were used as input. Inferred TF activities were integrated as a new Seurat assay, scaled, and averaged by annotated cell type and PCW. TF expressions and activities were visualized using bar plot and Seurat *FeaturePlot()*, respectively, and global regulatory patterns were summarized using heatmaps of the most variable TFs across cell types and PCW stages. TF-target regulatory networks were visualized using the igraph [https://igraph.org] and ggraph [https://ggraph.data-imaginist.com] R packages, with directed edges indicating activation or repression and force-directed layouts applied.

## Single-cell olfactory receptor (OR) dominance

OR dominance scores quantified the relative expression of the top-expressed OR versus the second-highest within each nucleus. Only expressed ORs were considered; cells without detectable OR expression received a score of zero. The score accounts for both relative difference and absolute expression of the top OR to mitigate sparsity. Scores were binned for distribution analysis across cell types. High-dominance cells were defined by dominance score >1, top OR expression ≥1, and ≤3 co-expressed ORs. Top dominant OR genes were identified per population. Analyses were performed using custom R scripts built on the Seurat framework.

## MERFISH spatial transcriptomics

MERFISH was performed using the Vizgen MERSCOPE platform. A custom 300-plex codebook was designed on the basis of the top cell type marker genes and the 57 most expressed ORs from the snRNA-seq 10x Genomics dataset. We evaluated this gene panel using the Merscope® Gene Panel Design Portal available at Vizgen (https://portal.vizgen.com). This resulted in the Merscope® 294 gene panel (Vizgen, BP0939 #10400002), and including 15 blank barcodes to serve as control for unspecific probes binding. Fresh frozen 14 μm coronal samples, cryosectioned using a CM3050 Leica cryostat, from the fronto-nasal region of one PCW9 and one PCW11 fetuses were placed on Merscope® slides (Vizgen, #10500102) and processed according to the manufacturer's user guides (Vizgen 91600002 Rev E and 91600112 Rev C), except for the fixation that was performed in 37% formalin for 1 h at RT. Cell boundary staining, gel embedding, clearing, and probe hybridization steps were performed without modification from the fresh-frozen sample preparation guide. Samples were imaged on the MERSCOPE instrument using the 300-plex imaging kit, and transcripts were decoded using the MERLIN pipeline provided by Vizgen. Three additional sections for each sample were stored in a 1.5 ml eppendorf tube at −80 °C prior to RNA extraction with miRNeasy microRNA kit (Qiagen, #217084). RNA quality was measured using an Agilent 2100 Bioanalyzer with the RNA 6000 Pico assay (Agilent, #5067-1513). The calculated DV200 sample PCW9 was 73% and its RIN was 7.7, for PCW9.2 the RIN score was 8.9, and for PCW11.5 the RIN score was 8.

## MERFISH data analysis

Tissue sections were processed individually using Vizgen Post-processing Tool (VPT, https://github.com/Vizgen/vizgen-postprocessing) after standard Merscope system analysis. Pre-processing of z-layer=2 ".tiff" images was done by local contrast adaptive histogram equalization (CLAHE) using custom parameters (clip_limit = 0.03, filter_size = 100 × 100) before cell segmentation using CellPose[65] algorithm based on "z = 2" nuclear DAPI and CellBound2 staining. VPT output files were then loaded into SpatialData[66] objects and processed following standard analysis. Briefly, for each Merscope slice, cells having fewer than 12 detected

transcripts were filtered out, transcripts per cells count matrices was then normalized and scaled. We then extracted the Main Olfactory region from the whole fetus head slice and export AnnData objects. We then integrate our 5 focused area using Harmony integration before UMAP dimension reduction and Leiden clustering. scMusketteers[38] algorithm was then used to transfer automatically labellings from our snRNA-seq Atlas to the 300 spatial target genes panel. Procedure was repeated 10 times and majority voting label per cell was then attributed to each spatial cell independently. Spatial domains analysis was done using the graph-based foundation model for spatial transcriptomics Novae[67] using pre-trained then fine-tuned available brain model using 7 spatial domains delineation.

To quantify the expression specificity of OR genes at the single-cell level, we computed a scaled dominance score for each cell (Dc) that expressed at least two OR transcripts. For each cell, let x1 and x2 represent the transcript counts of the most and second-most highly expressed OR genes, respectively. To penalize low-confidence cases with weak overall expression, we computed a log-scaled adjustment factor based on the top OR expression.

Dominance score (2)

$$D(c) = \left( \frac{x_1 - x_2}{x_1 + x_2 + \varepsilon} \right) \cdot \log(1 + x_1)$$

where $\varepsilon$ is a small constant ($\varepsilon = 10^{-9}$) included to ensure numerical stability.

## Multiplex fluorescent in situ hybridization

Two human fetuses at PCW9 were snap-frozen in liquid nitrogen and stored at −80 °C until use. Human tissues were cryosectioned using a CM3050 Leica cryostat at 16 μm. FISH was performed on frozen sections of the nasal regions by RNAscope Multiplex Fluorescent Kit v2 according to the manufacturer's protocol (Advanced Cell Diagnostics). Specific probes were used to detect *MUC5AC* (Cat N°: 312891), *GNRH1* (Cat N°: 562591) mRNAs. Hybridization with a probe against the Bacillus subtilis dihydrodipicolinate reductase (dapB) gene (Cat N°: 320871) was used as a negative control and 3-Plex Positive (Cat N°: 320861) as positive control.

## Immunofluorescence

One human fetus at PCW7.5, one at PCW9 and two at PCW12 were fixed in 4% Paraformaldehyde (PFA) in phosphate buffer (0.12 M, pH7.4), at 4 °C for 5 days. Specimens were rinsed and cold-stored in fresh 1X Phosphate Buffer Saline, then frozen in isopentane cooled by liquid nitrogen and stored at −80 °C until use. Human tissues were cryosectioned using a CM3050 Leica cryostat at 18 μm. Sections were thawed at room temperature and boiled at 80–90 °C in the citrate buffer (9 mL citric acid buffer 0.1 M + 41 mL sodium citrate buffer 0.1 M + water to 1 L) for the antigen retrieval. Slides were washed 3 times in PBS 1× and incubated for 3 days in the primary antibody solution (PBS 1×, 0.3% Triton X-100, 2% normal donkey serum) at 4 °C. The primary antibodies used were anti-Olig2 goat IgG (R&D Systems, #AF2418, diluted at 1/100), anti-Tubulin βIII, TUJ1, mouse monoclonal (Biolegend, #801201, diluted at 1/200), and anti-SOX2 (Y-7) goat polyclonal (Santa Cruz, # sc-17320, diluted at 1/200) and anti-Cytokeratin 5 (EP1601Y) rabbit monoclonal (Abcam, # ab52635, diluted at 1/500). Slides were rinsed 3 times in PBS 1x and incubated in the secondary antibody solution (PBS 1x, 0.3% Triton X-100, 2% normal donkey serum) for 1 h at room temperature. The secondary antibodies used were the Alexa Fluor 568 Donkey anti-Goat IgG 1:400 (ThermoFisher, # A-11057), Alexa Fluor 488 Donkey anti-Goat IgG 1:400 (ThermoFisher, #, 11055), Alexa Fluor 647-conjugated Donkey anti-Mouse 1:400 (ThermoFisher, # A-31571) and Alexa Fluor 568 Donkey anti-Rabbit IgG 1:400 (Thermo-Fisher, # A-10042). Finally, sections were cover slipped with

Fluoromount-G with DAPI (Invitrogen 00-4959-52), as an antifade mounting medium.

## Image analysis
Images were acquired on an inverted confocal microscope (Leica STELLARIS 5 microscope) and visualized using Leica Application Suite (LAS) X Office software (Imaging Core Facility of the University of Lille, France). Photoshop (version 26.0.0, Adobe Systems, San Jose, USA) was used to prepare the figures.

## Transmission electron microscopy
After collection, the PCW7.5 sample was rinsed and fixed *in toto* by immersion in 4% paraformaldehyde (PFA) and 0.2% glutaraldehyde prepared in 0.1 M phosphate buffer (pH 7.4) for 4 days at 4 °C. Following fixation, the sample was rinsed in 1× phosphate-buffered saline (PBS), and the head was sectioned into two coronal halves at the level of the frontal lobe. Samples were then post-fixed in 1% osmium tetroxide diluted in 0.1 M phosphate buffer for 30 min at room temperature, followed by extensive rinsing in 0.1 M PBS. Dehydration was performed through a graded ethanol series (50%, 70%, 95%, 95%, 100%, 100%, 100%), with each step lasting 1 min at room temperature. Samples were subsequently incubated in propylene oxide for 15 min, followed by a 10-min incubation in a propylene oxide/Araldite mixture. Embedding was carried out in Araldite resin for 48 h at 56 °C. Resin blocks were sectioned using a Leica UC7 ultramicrotome. Semi-thin sections (1 μm) were obtained, stained with 1% toluidine blue, and examined by light microscopy. Ultrathin sections (90 nm) were collected on nickel grids and contrasted with uranyl acetate and lead citrate. Observations were performed using a ZEISS EM900 transmission electron microscope operating at 80 kV.

## Statistics and reproducibility
Data are presented as mean ± standard error of the mean (SEM), unless otherwise stated. For all analyses, assumptions of normality and homogeneity of variance were assessed where appropriate. Differences between two groups were evaluated using the Wilcoxon rank-sum (Mann-Whitney $U$) test. Comparisons among three or more groups were performed using the Kruskal-Wallis's test, followed by post hoc pairwise comparisons where applicable. Lineage-specific expression dynamics were modeled using GAMs. Chi-square tests were used to quantify and visualize the number of genes from a given gene set expressed per cell. Differential gene expression and marker identification were performed using the MAST framework. A significance threshold of $\alpha = 0.05$ was applied for all statistical tests. The $p$ values and details of the statistical methods used for each experiment are provided in the figure legends or main text. Statistical analyses and data visualizations were conducted using R (version 4.5.1). Tissue samples were not randomized, and investigators were not blinded during sample collection, as no subjective measurements were involved.

## Reporting summary
Further information on research design is available in the Nature Portfolio Reporting Summary linked to this article.

## Data availability
All data generated or analyzed in the current study are included in the article and its Supplementary Figs. and Supplementary Data. The raw snRNA-seq datasets generated during the current study have been deposited in the European Genome/Phenome Archive (https://ega-archive.org/datasets/EGAD50000001712) and are available through controlled access due to privacy and consent restrictions related to human genomic data. Access to EGA archive datasets can be obtained by formal application to the Data Access Committee (DAC). Each DAC requires users/applicants to sign a Data Access Agreement (DAA), which details the terms and conditions of use for each dataset. An interactive Shiny application associated with processed human snRNA-seq data has been deposited on Zenodo (https://doi.org/10.5281/zenodo.18245692). MERFISH data have been deposited in Gene Expression Omnibus under accession code GSE303809. Source data are provided with this paper.

## Code availability
Custom scripts and codes used for snRNA-seq analysis can be found at https://github.com/ymbouamboua/HuDeCa. Python scripts for re-analysis and figures production of MERFISH experiments can be found at https://github.com/cobioda/human_fetal_olfactory_system.

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

## Acknowledgements

This work was supported by funding from INSERM HuDeca, INSERM cross-cutting program HuDeCA 2018 (to P.G. and P.B.), by a French Government Grant managed by the National Research Agency (ANR) under the action France 2030 with the reference ANR-24-CHBS-0002 (to P.G.), by ANR-19-CE14–0027 (to P.B.), ANR-23-IAHU-0007 (to P.B.), ANR-24-CE17-7692-01 (to P.B.), the Conseil Départemental des Alpes Maritimes (2016-294DGADSH-CV), the National Infrastructure France Génomique (Commissariat aux Grands Investissements) [ANR-10-INBS-09-03, ANR-10-INBS-09-02]; the 3IA@coted'azur [ANR-19-P3IA-0002], the PPIA 4D-OMICS [ANR-21-ESRE-0052], the Chan Zuckerberg Initiative [2017-175159-5022], and the IHU-Respirera [ANR-23-IAHU-0007] to P.B. Schemes and drawings were created with BioRender.com.

## Author contributions

P.B. and P.G. conceived and designed the study. Writing and preparation of the figures, P.B., Y.M., K.L. and P.G.; methodology and formal analysis, Y.M., K.L., S.N., M.C., C.A., L.C., M.-J.A. Data interpretation, conceptual discussions and critical manuscript revision: P.G., Y.M., K.L., S.N., V.P., P.B. Writing/review/editing: P.G., Y.M., K.L., S.N., V.P., P.B. Funding acquisition: P.B. and P.G.

## Competing interests

The authors declare no competing interests.
