## [Transparent Peer Review file · Nature Communications]

A Single-Cell and Spatial Atlas of Early Human Olfactory Development

Corresponding Author: Dr Paolo Giacobini

Version 0:

Reviewer comments:

Reviewer #1

(Remarks to the Author)

In this manuscript by Mbouamboua, Lebrigand et al., single-nucleus RNA sequencing and spatial transcriptomics were used to generate an atlas of the human fetal nasal region. Cell type composition changes in the olfactory (OE) and respiratory (RE) epithelia of human fetuses across post-conceptual weeks (PCW) 7 to 12 were explored. Multiplexed Error-Robust Fluorescence In Situ Hybridization (MERFISH) spatial transcriptomics integrated with snRNA-seq revealed changes in cell composition and gene expression over time. Similar to reports in mice, three maturation lineages were observed—neuronal, sustentacular, and microvillar—all three governed by differential transcription factor activities. Markers of olfactory sensory neuron development and pathways regulating epithelial patterning and OE morphogenesis were also identified.

The data reported in this manuscript constitute a very valuable resource in the field of olfaction and represent, to my knowledge, the first integrated molecular and spatial study of early human olfactory development.

However, I have some concerns (some of them critical) and suggestions that could improve the clarity and impact of the manuscript.

Main Concerns

1. Data Accessibility

To fulfill the resource purpose and ease data exploration, an interactive web application (using Shiny apps, for example) would be very useful.

2. Sample Demultiplexing Quality

The authors pool cells from different donors prior to snRNA-seq and then demultiplex the pools using Demuxafy. One pool contains samples from the two developmental extremes (PCW7 and PCW12). I am uncertain about the color coding in Extended Data Fig. 1c, but it appears that cells from S1-PCW7 are frequently misassigned to S6-PCW12. If correct, this is problematic and needs to be addressed.

3. Demultiplexing Concordance

In Extended Data Fig. 1b, the authors report similar numbers of assigned cells between SoupORcell and Vireo (two demultiplexing softwares implemented in Demuxafy) for each donor in each pool. What is the number of cells that are commonly assigned to each donor by both softwares? This concordance metric is crucial for assessing the reliability of the demultiplexing.

4. Unsupported Claims and Data-Text Mismatches

Many claims in the paper appear to be extrapolations, or at least not directly tested, and sometimes are not supported by the reported data. Here are a few specific examples:

a. Cell cycle quantification discrepancy

The text states (lines 196-199): "Quantification of cell cycle phase composition across developmental timepoints demonstrated a progressive increase in the proportion of post-mitotic cells, rising from ~25% at PCW7 to over 40% by PCW12 (Extended Data Fig. 4b)." However, the changes in the proportion of cycling and post-mitotic cells shown in

Extended Data Fig. 4b do not match these reported values.

b. mOSN maturation state

The authors report that, contrary to mice, GAP43 is expressed in human mOSNs (lines 291-293: "We found in this study that in human fetal olfactory neuroepithelium, GAP43, GNAL, RTN1, were expressed in both iOSNs and mOSNs" and Fig. 3c). However, GNG13, a marker of mature OSNs in mice, is expressed in only a few cells located at the tip of the UMAP (Fig. 3c). Moreover, the UMAP plot in Fig. 3b shows substantial overlap between the mOSN and iOSN clusters. While I understand that UMAP representations should be interpreted cautiously, this overlap raises doubts about whether the cells labeled as mOSNs actually correspond to mature neurons. Consistent with this concern, only PCW10 mOSNs show a marked increase in OR transcript levels compared to iOSNs (Fig. 6b).

I suggest projecting this dataset onto mouse data to determine the identity of these neurons. My concern is that what are termed iOSNs and mOSNs may actually represent immature neurons at two distinct maturation stages, and that this dataset contains very few truly mature OSNs characterized by GNG13 expression. These few mature cells may cluster with immature neurons due to their low abundance.

c. FGFR2 expression pattern

The text states that "FGFR2 is enriched in GBCs, INP and OSNs" (lines 380-381), but Extended Data Fig. 9 shows that only olfactory HBCs express this gene at high levels.

d. Transcription factor activity discrepancy

The text mentions that "non-neuronal compartments (SUS, basal cells) exhibited preferential activity of SIX1, NFKBIZ, and BARX2" (lines 415-416), but Fig. 4b shows that these transcription factors are predominantly active in INPs, iOSNs, and mOSNs. There also appears to be a mismatch between Fig. 4a and 4b for SIX1.

e. Dorso-ventral distribution analysis

Differences in the dorso-ventral distribution of OE and RE cell types between anterior and posterior sections are reported but not statistically tested. The relative sizes of OE and RE in both sections differ, which could account for the observed differences (Fig. 5j,k). I recommend quantifying this by normalizing the sizes of OE and RE in the anterior and posterior regions and directly comparing them.

5. Lineage Inference Methodology

The lineage inference in Fig. 3f,g is extracted from the UMAP representation. However, such representations are known to cause significant dataset distortions (see Chari and Pachter, 2023, <https://doi.org/10.1371/journal.pcbi.1011288>). I urge caution in interpreting this inference.

Why didn't the authors use PC space for this inference (as performed in the original paper by Fletcher et al., 2017, <https://doi.org/10.1016/j.stem.2017.04.003>), where much of the global data structure is preserved? Additionally, it is unclear how Slingshot parameter choices affected this inference. Did the authors force branching to 3 lineages? Were start and end points predefined? This information is missing from the methods section. I also suggest testing for dynamic differences between the three lineages across development and highlighting this in Fig. 3h.

6. OR Gene Expression Analysis - Critical Issue

Fig. 6c shows that almost 40% of mOSNs do not express an OR. This proportion is unusually high compared to rodent studies. Since the 3'UTR regions of OR genes are often misannotated, did the authors update the human GTF annotation file with re-annotation of OR 3'UTR regions prior to read mapping? If the default GTF file from Ensembl or UCSC was used, many OR genes were possibly missed, as only the 3' end of genes is sequenced in snRNA-seq. This has critical implications for all data reported in Fig. 6.

If re-mapping is performed, I also recommend standardizing the batch correction between snRNA-seq and MERFISH data using Harmony in both cases (see Luecken et al., 2022, <https://doi.org/10.1038/s41592-021-01336-8> and Antonsson and Melsted, 2024, <https://doi.org/10.1101/2024.03.19.585562>).

7. OR Expression Analysis Across Development

The adjusted dominance score used in the MERFISH data is a clever approach to assess monogenic OR gene expression. However, since the MERFISH data predominantly contains PCW9 samples and only one PCW11 section, developmental differences in OR expression dominance and magnitude cannot be tested. I suggest performing the same analysis on the snRNA-seq data, where this developmental comparison is feasible. I would also recommend performing the same analysis as in Fig. 6c while accounting for different developmental stages.

Minor Concerns

1. Readability: The manuscript is highly descriptive and occasionally tedious to read. Consider streamlining some sections.
2. Methods section completeness: The methods section needs improvement. Many analyses are omitted (e.g., sex difference analyses in Fig. 1e, analyses in Fig. 6), or descriptions do not match the main text (e.g., SCENIC is used for transcription factor activity inference in the text, but decoupleR is mentioned in the methods).
3. Figure annotations: Figure annotations should be improved to clearly reflect what is measured. For example, the scale bar meaning in Extended Data Fig. 1c is unclear, and the y-axes of Fig. 5j,k are not labeled. Extended Data Fig. 5b is also missing a y-axis label.

4. Data visualization: It is difficult to link cluster attributions with expression heatmaps displayed on UMAP plots (e.g., Figures 3c and 4a, Extended Data Figures 2b and 4c). To better highlight cell type specificity, I suggest using violin or box plots instead.
5. Clustering anomaly: I am curious why mOSNs cluster with respiratory ciliated cells rather than with iOSNs and other neurons in Extended Data Fig. 5a. Please clarify.
6. Data source clarity: In the main text, please mention that results shown in Fig. 6 come from snRNA-seq data, since this figure follows results from MERFISH data.
7. Missing figure: Fig. 7f is mentioned in the manuscript but is missing.
8. Lack of quantification: Images are often shown but not quantified (e.g., Fig. 2e-g and Fig. 7e). Providing quantifications would substantiate the claims made.

Reviewer #2

(Remarks to the Author)

Summary:

Mbouamboua et al report efforts to develop a single cell and spatial transcriptomic atlas from embryonic human olfactory and respiratory nasal mucosa. Tissue was obtained from 10 human fetuses between 7 and 12 weeks gestation, used for snRNA-seq and MERFISH. This will be a new resource of interest, as present data sets from human are from adult olfactory biopsies or from animal models. A novel and important finding is the identification of maturing OSNs present even in first trimester, with singular OR expression patterning emerging early. While there are many strengths, some concerns are outlined below, especially involving the computational analysis approaches and associated interpretations. Because this is a descriptive resource, it is important to avoid attempting to over-state some conclusions, such as lineage trajectory analyses which are only limited models based on static sampling and annotations.

Specific comments:

Evidence suggesting that olfactory HBCs may be present in early human OE development is of interest. If correct, this does seem to contrast mechanisms identified in murine olfactory development, which suggest that HBCs only emerge just before birth, well after the neuroepithelium and patterning have been generated from more GBC-like basal and apical progenitors. Higher magnification staining confirming HBC markers present in regions containing neurogenic cells would be very helpful. The discrimination between bona fide olfactory HBCs and adjacent respiratory basal cells in scRNA-seq data sets containing both populations can be challenging, so careful annotation of these populations is important. It is not clear that several of the markers selected here are well-established (COL17A1, RASSF6, NNAT), so some validation would be important.

The lncRNA MEG3 has been reported to be enriched in human olfactory HBCs compared to respiratory basal cells. Is this the case in these embryonic samples? If so, it may help to distinguish these populations.

Many of the canonical basic HLH TFs used to identify GBCs and INPs (ASCL1, NEUROG1, NEUROD1) seem to not be used here. Are these transcripts not present embryonically, or is there a reason they are not utilized?

Several core analyses treat cells as the replicate rather than donors, which risks pseudoreplication and stage-donor confounding, especially at early stages where there is only a single donor (PCW7–9).

The authors should describe how maternal contamination was assessed and excluded. At minimum, show that sex chromosome expression from the subjects is consistent with fetal sex (or inferred fetal sex), and that erythroid contamination is negligible. This is important because, while these early embryos will have immune cells of their own (particularly myeloid-lineage cells), some immune cells could be maternal.

The authors state that Slingshot was run on UMAP coordinates. Because UMAP is a nonlinear visualization method and distances between cells are not preserved, it is not an appropriate space for trajectory inference. The analysis should be re-run on PCA, and the results plotted on UMAP only for display. This would ensure that inferred developmental trajectories reflect transcriptomic structure rather than visualization artifacts.

At PCW10 the authors highlight enrichment of hypoxia/stress/mitochondrial programs. These signatures are highly sensitive to cold ischemia and processing conditions in snRNA seq. The way to demonstrate biological, stage linked signal is to re-test these enrichments using donor level statistics (pseudo bulk or mixed effects), including stage as a fixed effect and donor as a random effect, and adding processing covariates where available (e.g., time to freezing). If the signal persists under those models, the claim is much stronger.

Claims about changing cell type proportions (e.g., endothelial/lymphatic expansion, progenitor decline) likewise need donor aware compositional modeling to control for anatomical sampling.

For cell-cell communication, the CellChat results probably reflect composition rather than biology. A more robust approach is to compute networks per donor (or balance donor groups), and report only ligand-receptor pathways that replicate across donors with FDR control and minimum cell count thresholds.

For GO analysis, current best practices for snRNA seq favor performing enrichment on donor aware differential signals derived from a GLM (pseudo bulk) or mixed effects framework, rather than on pooled cell level lists. Suggest that you re-run GO using such models and report FDR along with leading edge genes.

The MERFISH panel included 57 ORs on coronal sections. The Discussion (line 647) states that there is evidence of a DV expression gradient or pattern of OR expression, yet the data do not clearly show this. Is it possible to better support this important finding.

Minor

In Seurat, `NormalizeData` uses `LogNormalize`, while `FindVariableFeatures(method="vst")` selects HVGs, but the current text conflates these.

The cell cycle section needs reconciliation. The text says “non cycling cells were rare,” yet the fraction of post mitotic cells rises from ~25% at PCW7 to >40% at PCW12, which is not rare. Consider updating the language to reflect the observed increase in post mitotic fractions with age.

Reviewer #3

(Remarks to the Author)

This study integrates single-nucleus profiling with high-resolution spatial transcriptomics to chart cellular composition and organization of the human nasal and olfactory epithelium across PCW7-12. The authors catalog diverse epithelial, neuronal, glial, stromal, vascular, and immune populations, reconstruct putative developmental trajectories within the olfactory epithelium, and nominate candidate regulators and signaling interactions. The spatial datasets are particularly informative, defining coherent epithelial domains, revealing anterior-posterior and dorsal-ventral patterning, and localizing lineage markers in situ. Notably, spatially resolved maps of olfactory receptor transcripts identify receptor-enriched domains and support an early emergence of largely single-receptor expression per neuron, providing a concrete framework for studying human olfactory morphogenesis and receptor gene choice in native tissue.

This is a strong resource that will be useful to the community. I find the one neuron - one receptor observation particularly interesting; while consistent with mouse, establishing it directly in human fetal tissue is essential and, in my view, constitutes a discovery rather than mere validation, since this is the first systematic analysis in human. The study also leverages spatial transcriptomics effectively to anchor cell identities and gene expression in anatomy. That said, the spatial resolution of receptor usage could go further, especially in distinguishing spatial distribution patterns among different ORs.

One main criticism is that the manuscript is substantially over length for Nature Communications (limit ~5000 words for the main text; current draft ~7800) and the narrative loses focus as a result. In particular, the narration of GO terms from ED Fig 3 taking up the entire page 6 adds little, and the cell-cell communication panels in ED Fig 7, 8, 9 are not compelling without validation. Walking through these figures panel-by-panel in the main text feels redundant and dilutes the story. I suggest substantially cutting the main text, removing scattered minor observations, and tightening to a few unifying take-home insights. This would keep attention on the strongest contributions, namely the spatial organization of OE versus RE and the early emergence of predominantly single-OR expression in human.

Other specific concerns that the authors need to address in a revision include:

1. Provide experimental validation for the cell-cell communication analysis. At minimum, show in situ co-localization for some of the prioritized axis (for example SLIT-ROBO or SEMA3A-NRP1/PLXNA4).
2. The current approach for the GO analysis in ED Fig 3 is effectively pseudo-bulk gene expression across mixed cell types, so changes in cell type proportions likely drive the top terms rather than within-cell-type transcriptional changes. It would be more informative to perform differential analyses within stable cell types.
3. ED Fig. 3c highlights hypoxia stress signal. Is this expected biologically, or could it reflect pre-analytical variables (reasons of pregnancy termination, ischemic interval, sample processing delay)?
4. The authors note a built-in CellPose segmentation for MERFISH but did not show how well it performed. Epithelia are very high density and hard to segment. Please include DAPI and a cell boundary marker overlaid with the final cell masks. This is essential, since precise segmentation underpins all single-cell analyses and is critical for evaluating the 1 neuron - 1 receptor principle.
5. In regulon results in Fig 4, several factors show a pronounced mismatch between expression and inferred activity (e.g., BARX2, NFKBIZ). How should this be interpreted?
6. In Fig 5j-k, the curves for spatial distribution of cell types and gene expression are very hard to read. Is there a better way to plot this so the patterns are clearer?
7. The dominance score in Fig 7 is a very good idea. But in Fig 7c-d the high-dominance OSNs look sparse. Is this very low abundance consistent with the scRNAseq single-OR rate? Could MERFISH sensitivity drop off and under-detect OR transcripts? Or could it be related to limitations of MERFISH gene panels only includes 57 OR genes?
8. The text references a Fig 7f panel that is not present. Is data omitted by mistake? Please restore the intended panel.
9. Beyond what is shown in Fig 7, it would be very interesting to look into whether individual ORs show spatial patterning. For example, specificity or gradients along the dorsal-ventral axis. Demonstrating this would showcase the unique advantages of spatial transcriptomics to reveal insights that other methods cannot, and would further strengthen the biological significance of the study.

Version 1:

Reviewer comments:

Reviewer #1

(Remarks to the Author)

The authors adequately answered all my questions (data availability, multiplexing quality, concordance, cell cycle quantification, renaming iOSNs, FGFR2, TF analysis, umap to PCA, lineage distribution, adjusted dominant scores, removal

of repeats).

I have no additional questions.

(Remarks on code availability)

The code seems appropriate and well annotated.

Reviewer #2

(Remarks to the Author)

In general, the authors were highly responsive to my comments and those of the other Reviewers. As such, the revised manuscript is strengthened. Many of the problems from the original snRNA-seq analysis have now been corrected. The manuscript text has been edited to reduce lengthy descriptive commentary, improving readability. I have only minor concerns, not warranting re-review.

Some of the conclusions on olfactory epithelial and olfactory receptor expression patterns from MERFISH remain overstated, because only 2 donor specimens are analyzed spatially (Extended Data Figure 11a) and only a small number of OR genes are visualized, with limited statistical analysis feasible, so this should be stated.

This work should be a valuable resource and the authors should be commended for their efforts.

(Remarks on code availability)

Code is well annotated, permits reanalysis if needed

Reviewer #3

(Remarks to the Author)

The authors have addressed my main concerns in a thoughtful and largely effective manner. The revised manuscript is substantially improved in writing and narrative focus, and the overall story now reads much more cleanly.

First, the authors streamlined the manuscript by removing several sections and figures that previously diluted the core message. In particular, the prior extended analyses that I felt were not compelling without validation (including the cell-cell communication and GO term focused content) have been removed, which substantially improves readability and keeps attention on the strongest contributions of the study. Relatedly, the computational framework is improved, including trajectory inference, which strengthens the developmental interpretation.

Second, the revision adds several pieces of new data and analyses that improve the manuscript. The new immunostaining in Fig 2j provides helpful validation and clearer anatomical grounding. The new analysis of dorso ventral distribution in the spatial data (ED Fig 8) is also interesting and showcases the unique advantages of spatial transcriptomics beyond cell type annotation.

Third, I appreciate the added quality control for MERFISH segmentation. The overlay of cell boundary staining and CellPose segmentation masks looks convincing.

One remaining point is that Fig 5j remains difficult to interpret. While I understand that some improvement was made, the figure is still visually dense and the key spatial patterns are not easy to extract. I encourage the authors to consider an alternative visualization that makes the trends more immediately readable (for example, splitting into separate panels or using a heatmap or binned density representation).

Overall, I feel the authors have done an appropriate revision and the manuscript is substantially strengthened and fitting for publication.

(Remarks on code availability)

COMMENTS TO THE AUTHOR:**Reviewer #1 :**

The data reported in this manuscript constitute a very valuable resource in the field of olfaction and represent, to my knowledge, the first integrated molecular and spatial study of early human olfactory development.

However, I have some concerns (some of them critical) and suggestions that could improve the clarity and impact of the manuscript.

Major comments:**1. Data Accessibility**

To fulfill the resource purpose and ease data exploration, an interactive web application (using Shiny apps, for example) would be very useful.

We thank the reviewer for this valuable suggestion. To facilitate interactive exploration of the spatial transcriptomics (ST) data, all ST datasets have been submitted to GEO as Xenium Explorer archives. In addition, responding to the reviewer's recommendation, we have developed a Shiny-based interactive web application. The Shiny application, together with all processed data, has been deposited on Zenodo. This substantially enhances data accessibility and usability for the community.

2. Sample Demultiplexing Quality

The authors pool cells from different donors prior to snRNA-seq and then demultiplex the pools using Demuxafy. One pool contains samples from the two developmental extremes (PCW7 and PCW12). I am uncertain about the color coding in Extended Data Fig. 1c, but it appears that cells from S1-PCW7 are frequently misassigned to S6-PCW12. If correct, this is problematic and needs to be addressed.

Because donor misassignment can arise from insufficient quality control of bulk RNA-derived genotype references, we reprocessed all donor VCFs using a unified and more stringent pipeline.

Specifically, per-donor VCFs were merged using bcftools merge and filtered using a minimum allele count (MAC) ≥ 1 and F_MISSING < 0.2 , consistent with the thresholds applied in our bulk RNA-seq demultiplexing workflow. We then evaluated SNP- and donor-level quality metrics, including overall missingness, the proportion of variants with $>50\%$ missing data, and the number of SNPs with minor allele frequency (MAF) ≥ 0.05 . All metrics confirmed that the final bulk genotype reference was of high completeness and met established quality control standards.

Using this curated SNP set, we repeated genotype concordance analyses and computed Pearson correlations of bulk RNA expression profiles across donors. Both analyses showed clear donor-specific separation, with no evidence of increased similarity between S1-PCW7 and S6-PCW12 that would indicate donor mixing or misassignment. Therefore, the apparent similarity observed in the original version of Extended Data Fig. 1c did not reflect errors in donor identity or limitations in genotype quality.

The revised analysis and updated figure now accurately reflect donor-specific transcriptomic profiles and confirm robust demultiplexing performance across pooled samples.

3. Demultiplexing Concordance

In Extended Data Fig. 1b, the authors report similar numbers of assigned cells between Souporcell and Vireo (two demultiplexing softwares implemented in Demuxafy) for each donor in each pool. What is the number of cells that are commonly assigned to each donor by both softwares? This concordance metric is crucial for assessing the reliability of the demultiplexing.

Response: Using the same stringent framework applied throughout our demultiplexing analyses, we quantified donor-wise concordance between Vireo and Souporcell assignments using bulk RNA-derived genotype references.

For each pool, inferred donors were mapped to bulk samples based on allele concordance (minimum concordance ≥ 0.60 ; ambiguity delta ≤ 0.05). Cells were retained as high-confidence singlets only when donor assignments from both tools were concordant and matched the same bulk donor; all other cells were excluded as ambiguous or doublets.

Using this definition, we computed the number and proportion of cells commonly assigned to each donor by both methods. These concordant singlets constitute the final dataset used for downstream analyses and provide a conservative measure of demultiplexing reliability. Donor-wise concordance counts are now reported in Extended Data Fig. 1c, d, and the full implementation is available in our GitHub repository (<https://github.com/ymbouamboua/HuDeCa/tree/main>).

4. Unsupported Claims and Data-Text Mismatches

Many claims in the paper appear to be extrapolations, or at least not directly tested, and sometimes are not supported by the reported data. Here are a few specific examples:

a. Cell cycle quantification discrepancy

The text states (lines 196-199): "Quantification of cell cycle phase composition across developmental timepoints demonstrated a progressive increase in the proportion of post-mitotic cells, rising from ~25% at PCW7 to over 40% by PCW12 (Extended Data Fig. 4b)." However, the changes in the proportion of cycling and post-mitotic cells shown in Extended Data Fig. 4b do not match these reported values.

Response: we thank reviewer for identifying this discrepancy. In the submitted version, G1-phase cells were incorrectly annotated as post-mitotic, for which we do apologize. This annotation error has now been corrected. In the revised analysis of the main olfactory epithelium, quantification across developmental time showed a progressive decline in actively cycling (G2/M) cells between PCW7 and PCW12 (~35% to ~16%), accompanied by an increase in G1 cells (~12% to ~40%), consistent with a transition from proliferation to differentiation (new Extended Data Fig. 6b). S-phase, post-mitotic, and non-cycling populations showed only modest temporal variation (new Extended Data Fig. 6b).

b. mOSN maturation state

The authors report that, contrary to mice, *GAP43* is expressed in human mOSNs (lines 291-293: "We found in this study that in human fetal olfactory neuroepithelium, *GAP43*, *GNAL*, *RTN1*, were expressed in both iOSNs and mOSNs" and Fig. 3c). However, *GNG13*, a marker of mature OSNs in mice, is expressed in only a few cells located at the tip of the UMAP (Fig. 3c). Moreover, the UMAP plot in Fig. 3b shows substantial overlap between the mOSN and iOSN clusters. While I understand that UMAP representations should be interpreted cautiously, this overlap raises doubts about whether the cells labeled as mOSNs actually correspond to mature neurons. Consistent with this concern, only PCW10 mOSNs show a marked increase in OR transcript levels compared to iOSNs (Fig. 6b).

I suggest projecting this dataset onto mouse data to determine the identity of these neurons. My concern is that what are termed iOSNs and mOSNs may actually represent immature neurons at two distinct maturation stages, and that this dataset contains very few truly mature OSNs characterized by *GNG13* expression. These few mature cells may cluster with immature neurons due to their low abundance.

Response: We thank the reviewer for this insightful and constructive comment. Based on additional analyses and a careful reassessment of canonical marker expression, we agree that the original annotations overstated the presence of fully mature olfactory sensory neurons (mOSNs) in this dataset. We have therefore revised our cell-type annotations to more accurately reflect the neuronal maturation continuum captured in the sampled developmental window.

Specifically, cells previously annotated as INP, iOSN, and mOSN are now classified as INP and iOSN only. Cells formerly designated as mOSNs most likely represent late immature neurons rather than fully mature OSNs. This interpretation is supported by the very low abundance of *GNG13*-positive cells, which do not form a distinct cluster and are restricted to a small subset of nuclei at the periphery of the UMAP. Consistent with this, only few PCW10 cells show a marked increase in odorant receptor transcript levels, indicating that truly mature OSNs are rare at the fetal stages analyzed.

To validate this revised annotation, we examined the expression patterns of canonical olfactory epithelium markers described in mouse studies (e.g., Fletcher et al., 2017). In our data, cycling progenitors express *TOP2A*, *MKI67*, and *ASCL1*; immediate neuronal progenitors express *NEUROG1* and *NEUROD1* with variable *GAP43*; and immature OSNs are characterized by high *GAP43* and *GNG8* expression. These marker distributions are recapitulated in our dataset and support a progressive maturation continuum rather than discrete immature and mature OSN populations.

We also considered the reviewer's suggestion to project the human dataset onto mouse olfactory epithelium references. However, given species-specific differences in developmental timing, as shown in this study, transcriptional dynamics, and marker gene usage in the olfactory system, direct cross-species projection at these early fetal stages may be difficult to interpret and could introduce ambiguity rather than clarify neuronal identity. Instead, we relied on conserved marker hierarchies and internal maturation signatures, such as

cell-cycle exit, neurogenic markers, *GAP43/GNG8* expression, and odorant receptor induction, to reassess neuronal identity within the human dataset itself.

Together, these results indicate that the majority of neuronal cells captured in this study correspond to immature or early differentiating neurons. Fully mature OSNs, as defined by robust *GNG13* expression, are exceedingly rare during the developmental stages analyzed and therefore do not segregate as a distinct cluster. The revised annotations, updated figures, and accompanying dot plots now accurately reflect this developmental continuum and address the reviewer’s concerns.

Notably, we observe a progressive increase in odorant receptor (OR) expression along the differentiation trajectory from GBCs to INPs and subsequently to iOSNs. This pattern indicates that odorant receptor expression is gradually established during neuronal differentiation, increasing as cells differentiate from progenitors to immature olfactory sensory neurons.

The revised analysis and updated figure now accurately reflect the correct annotations.

c. *FGFR2* expression pattern

The text states that "*FGFR2* is enriched in GBCs, INP and OSNs" (lines 380-381), but Extended Data Fig. 9 shows that only olfactory HBCs express this gene at high levels.

Response: We thank the reviewer for identifying this inconsistency between the text and the Figure. We recognize that the statement regarding *FGFR2* enrichment was not sufficiently supported by the data. In the

revised version of the manuscript, we have removed the former Extended Data Fig. 9, as requested by Reviewer 3 who considered Extended Data Figs. 3, 7, 8, and 9 not sufficiently compelling. We have thus substantially revised and streamlined the main text and removed those Figures from the article to sharpen the narrative focus. As a consequence, FGFR2 expression is no longer presented or discussed in the revised manuscript.

d. Transcription factor activity discrepancy

The text mentions that "non-neuronal compartments (SUS, basal cells) exhibited preferential activity of SIX1, NFKBIZ, and BARX2" (lines 415-416), but Fig. 4b shows that these transcription factors are predominantly active in INPs, iOSNs, and mOSNs. There also appears to be a mismatch between Fig. 4a and 4b for SIX1.

Response: We thank the reviewer for identifying this discrepancy. The inconsistency arose from an error in the text rather than from the underlying data or analyses. The transcription factor activity inference shown correctly reflects predominant activity of SIX1, NFKBIZ, and BARX2 in neuronal progenitors and immature neurons. In addition, following the revision of our neuronal annotations, we re-analyzed transcription factor (TF) activities and revised Fig.4 accordingly.

e. Dorso-ventral distribution analysis

Differences in the dorso-ventral distribution of OE and RE cell types between anterior and posterior sections are reported but not statistically tested. The relative sizes of OE and RE in both sections differ, which could account for the observed differences (Fig. 5j, k). I recommend quantifying this by normalizing the sizes of OE and RE in the anterior and posterior regions and directly comparing them.

Response: Our intent in presenting the anterior and posterior OE/RE spatial cell-type organization was to provide a precise view of epithelial cell-type architecture. However, the large difference in tissue length (i.e., OE is ~200 μm anteriorly versus ~1200 μm posteriorly, with the opposite pattern in the RE) precludes a direct side-by-side comparison. To address this, we now provide in Extended Data Figure 8 (related to the spatial transcriptomics data), which offers additional comparative information and allows readers to better appreciate the organization and differences between anterior and posterior sections.

The image below show spatial transcriptomic maps showing annotated cell types along the left and right olfactory epithelial lamellae for PCW9-A-S1 (anterior section 1), PCW9-A-S2 (anterior section 2), PCW9-P-S1 (posterior section 1), and PCW9-P-S2 (posterior section 2) specimens. Each point represents a segmented cell colored by its predicted cell type of the OE. Below each spatial map, heatmaps display the normalized proportion of each annotated cell type across spatial bins (50 μm bins), revealing conserved epithelial layering and sample-specific variation in cell-type composition.

5. Lineage Inference Methodology

The lineage inference in Fig. 3f, g is extracted from the UMAP representation. However, such representations are known to cause significant dataset distortions (see Chari and Pachter, 2023, <https://doi.org/10.1371/journal.pcbi.1011288>). I urge caution in interpreting this inference.

Why didn't the authors use PC space for this inference (as performed in the original paper by Fletcher et al., 2017, <https://doi.org/10.1016/j.stem.2017.04.003>), where much of the global data structure is preserved? Additionally, it is unclear how Slingshot parameter choices affected this inference. Did the authors force branching to 3 lineages? Were start and end points predefined? This information is missing from the methods section. I also suggest testing for dynamic differences between the three lineages across development and highlighting this in Fig. 3h.

Response: We agree that UMAP-based representations can distort global structure and should be interpreted with caution for lineage inference.

In response, we have re-performed the entire Slingshot lineage inference exclusively in principal component (PC) space, which preserves the global data manifold, in line with the recommendations of Chari and Pachter (2023) and consistent with the approach used by Fletcher et al. (2017). All lineage topology, pseudotime estimation, and branching structure are now derived solely from PCA space. UMAP is used only for visualization of the inferred trajectories for clarity and is not used in the computation of lineages or pseudotime.

Regarding Slingshot parameterization, no branching structure was manually enforced. The algorithm consistently identified three lineages in an unsupervised manner. Based on established biological knowledge of olfactory epithelial differentiation, olfactory horizontal basal cells (OHBCs) were specified as the root population, while immature OSNs (iOSNs), microvillar (MV) cells, and sustentacular (SUS) cells were defined as terminal populations. These choices reflect known differentiation directionality and were not used to constrain lineage number or topology.

We have now explicitly documented all Slingshot parameters, root and terminal definitions, and the use of PCA space in the Methods section.

In addition, we tested for dynamic differences across the three inferred lineages over development and have updated Fig. 3h to highlight lineage-specific pseudotime progression. These revisions strengthen the robustness and interpretability of the lineage analysis and directly address the reviewer's concerns.

6. OR Gene Expression Analysis - Critical Issue

Fig. 6c shows that almost 40% of mOSNs do not express an OR. This proportion is unusually high compared to rodent studies.

Response: We thank the reviewer for raising this point. While the proportion of neurons lacking detectable OR expression appears high when compared to rodent datasets, it is consistent with published human data. In particular, a previous human olfactory epithelium study reported that approximately 40% of immature OSNs do not exhibit detectable OR transcripts (Durante et al., Nature Neuroscience, 2020; their Fig. 3b).

That said, as discussed in our response to point 4b, based on additional analyses and a careful reassessment of canonical marker expression, we agree that the original annotations overstated the presence of fully mature olfactory sensory neurons (mOSNs) in this dataset. We have therefore revised our cell-type annotations to more accurately reflect the neuronal maturation continuum present during the sampled developmental window. Specifically, cells previously annotated as INP, iOSN, and mOSN are now classified as INP and iOSN only.

The revised data are presented in the updated Fig. 5d. Under this revised annotation, we observe that approximately 40% of iOSNs do not express ORs, more than 40% express a single OR, ~10% express two ORs, and only a small fraction expresses three or more ORs. This distribution is consistent with the progressive establishment of the one-neuron–one-receptor rule during human development, well before birth.

The Results and Discussion sections have been revised accordingly to reflect this updated interpretation.

Durante et al, 2020 (adult human OE)

Our dataset (fetal human OE)

Since the 3'UTR regions of OR genes are often misannotated, did the authors update the human GTF annotation file with re-annotation of OR 3'UTR regions prior to read mapping? If the default GTF file from Ensembl or UCSC was used, many OR genes were possibly missed, as only the 3' end of genes is sequenced in snRNA-seq. This has critical implications for all data reported in Fig. 6.

Response: we thank the reviewer for raising this important point regarding 3'UTR annotation of olfactory receptor (OR) genes. Read mapping and quantification were performed using the standard human reference annotation (hg38/Ensembl), without manual extension of OR 3'UTRs.

To directly assess the potential impact of incomplete 3'UTR annotation, we performed a systematic inspection of aligned BAM files at OR loci. We quantified the proportion of reads mapping within the annotated OR gene body ("inside") and in downstream windows extending 100 nt, 200 nt, and 500 nt beyond the annotated 3' end. Across all OR loci, extending the annotation by 100 nt would recover an additional ~1.97% of OR-associated signal relative to the signal already present in our count matrices, while a 500 nt extension would recover ~10.85% additional signal. These values represent global estimates and vary across individual OR genes.

While extending 3'UTR annotations could modestly increase OR detection sensitivity, especially for specific loci, larger extensions substantially increase the risk of overlapping neighbouring genes in this gene-dense genomic context, potentially introducing ambiguity in read assignment. Importantly, the majority of OR-derived signal is already captured within the annotated loci, and relative comparisons across cell types and developmental stages, which form the basis of Fig. 6, are therefore unlikely to be qualitatively affected.

Together, these analyses indicate that although incomplete 3'UTR annotation may lead to a modest underestimation of OR expression for some genes, it does not materially impact the main conclusions regarding OR expression dynamics and maturation patterns reported in Fig. 6.

		extend_100nt	extend_200nt	extend_500nt
Somme	126382	2491	3934	13714
%age missed		1,97	3,11	10,85

If re-mapping is performed, I also recommend standardizing the batch correction between snRNA-seq and MERFISH data using Harmony in both cases (see Luecken et al., 2022, <https://doi.org/10.1038/s41592-021-01336-8> and Antonsson and Melsted, 2024, <https://doi.org/10.1101/2024.03.19.585562>).

Response: we thank the reviewer for this constructive suggestion. Although we did not perform re-mapping of raw reads, we re-integrated the snRNA-seq data using Harmony to correct for batch effects across donors and technical replicates, following current best practices. This integration improved cross-sample alignment while preserving known biological structure. Integration of the MERFISH data was likewise performed using Harmony, ensuring a consistent batch-correction framework across modalities.

7. OR Expression Analysis Across Development

The adjusted dominance score used in the MERFISH data is a clever approach to assess monogenic OR gene expression. However, since the MERFISH data predominantly contains PCW9 samples and only one PCW11 section, developmental differences in OR expression dominance and magnitude cannot be tested. I suggest performing the same analysis on the snRNA-seq data, where this developmental comparison is feasible. I would also recommend performing the same analysis as in Fig. 6c while accounting for different developmental stages.

Response: we thank reviewer for this suggestion. We implemented an OR dominance score analysis in the snRNA-seq dataset, which spans multiple developmental stages (PCW7-PCW12), using the same adjusted dominance metric used for MERFISH. Because snRNA-seq captures OR transcripts sparsely, the dominance score distribution is markedly compressed (almost all nuclei <2), whereas MERFISH supports values >3 . Within this snRNA-seq-specific dynamic range, we observe the expected developmental progression: low-dominance bins (<1) are enriched for INP and other epithelial cells, whereas iOSN dominate the higher-dominance bins. When applying an appropriate threshold for snRNA-seq (adjusted score >1 , top OR expression ≥ 1 UMI, and ≤ 3 expressed ORs), the majority of high-dominance nuclei correspond to iOSN (142 iOSN vs 16 INP and 18 Others). These results closely mirror the MERFISH trend, confirming that monogenic OR acquisition increases with neuronal maturation across both datasets.

Revised Fig. 6h, i

Minor Concerns

1. Readability: The manuscript is highly descriptive and occasionally tedious to read. Consider streamlining some sections.

Response: We agree with the reviewer that the manuscript was overly descriptive in places and that this reduced overall readability. In line with this comment, and consistent with the suggestions of Reviewer 3 to focus more strongly on the central message, we have substantially streamlined the text. Specifically, we removed the detailed descriptions of the former Extended Data Figs. 3, 7, 8, and 9, which interrupted the narrative flow without materially advancing the main conclusions. These revisions improve clarity and readability while keeping the focus on the key biological insights.

2. Methods section completeness: The methods section needs improvement. Many analyses are omitted (e.g., sex difference analyses in Fig. 1e, analyses in Fig. 6), or descriptions do not match the main text (e.g., SCENIC is used for transcription factor activity inference in the text, but decoupleR is mentioned in the methods).

Response: We have substantially revised the Methods section to ensure that all analyses described in the main text are fully documented and consistent. Specifically, we (i) added detailed descriptions of the sex-difference analysis shown in Fig. 1e, (ii) included the complete computational workflow for analyses presented in Fig. 6, and (iii) corrected the description of transcription-factor activity inference. The main text previously referenced SCENIC was a mistake; all TF activity analyses were performed using decoupleR (as now stated consistently throughout the manuscript). The revised Methods now explicitly list all software, versions, parameters, and processing steps.

For clarity, we also expanded the figure legend for Fig. 1e to specify that the values shown represent residualized mean UMI counts per cell type after regression on developmental stage (post-conception weeks, PCW). Boxes indicate distributions by sex, and points represent individual cells colored by PCW.

3. Figure annotations: Figure annotations should be improved to clearly reflect what is measured. For example, the scale bar meaning in Extended Data Fig. 1c is unclear, and the y-axes of Fig. 5j, k are not labeled. Extended Data Fig. 5b is also missing a y-axis label.

Response: We have revised all figures accordingly, adding missing axis labels, clarifying the scale bar in Extended Data Fig. 1c, and correcting the y-axes of Fig. 5j, 5k, and Extended Data Fig. 5b. All figure annotations have now been updated for clarity and accuracy. A scale bar for Fig. 5j, k has been added to the revised Figure.

4. Data visualization: It is difficult to link cluster attributions with expression heatmaps displayed on UMAP plots (e.g., Figures 3c and 4a, Extended Data Figures 2b and 4c). To better highlight cell type specificity, I suggest using violin or box plots instead.

Response: In the revised manuscript, we have added violin plots to display expression levels of key marker genes across annotated cell clusters.

5. Clustering anomaly: I am curious why mOSNs cluster with respiratory ciliated cells rather than with iOSNs and other neurons in Extended Data Fig. 5a. Please clarify.

Response: The apparent clustering of mOSNs with respiratory ciliated cells in the original Extended Data Fig. 5a was due to residual technical artifacts (ambient RNA and sex-linked signal) and incomplete batch correction in the initial integration. We re-processed the data by (1) removing ambient RNA contamination, (2) filtering sex-linked contamination, (3) re-running batch integration using Harmony, and (4) re-clustering. After these steps the cluster-correlation heatmap now shows per example iOSN associating with INP and other neuronal populations (updated Extended Data Fig. 5a). We have replaced the figure and updated a Methods paragraph describing the processing workflow.

6. Data source clarity: In the main text, please mention that results shown in Fig. 6 come from snRNA-seq data, since this figure follows results from MERFISH data.

Response: The text has been revised following the reviewer's recommendation.

7. Missing figure: Fig. 7f is mentioned in the manuscript but is missing.

Response: This has been corrected.

8. Lack of quantification: Images are often shown but not quantified (e.g., Fig. 2e-g and Fig. 7e). Providing quantifications would substantiate the claims made.

Response: We agree that quantitative analyses would strengthen these observations. However, robust quantification of the indicated stainings would require systematic sectioning and staining of multiple independent fetal specimens, with sampling across the entire main olfactory epithelium to account for regional variability. Given the limited availability of human fetal tissue and the scope of the present study, such analyses were not feasible.

To avoid overinterpretation, we have therefore revised the text to ensure that these data are presented as qualitative observations rather than quantitative conclusions.

Reviewer #2:

This will be a new resource of interest, as present data sets from human are from adult olfactory biopsies or from animal models. A novel and important finding is the identification of maturing OSNs present even in first trimester, with singular OR expression patterning emerging early. While there are many strengths, some concerns are outlined below, especially involving the computational analysis approaches and associated interpretations. Because this is a descriptive resource, it is important to avoid attempting to over-state some conclusions, such as lineage trajectory analyses which are only limited models based on static sampling and annotations.

Major comments:

Evidence suggesting that olfactory HBCs may be present in early human OE development is of interest. If correct, this does seem to contrast mechanisms identified in murine olfactory development, which suggest that HBCs only emerge just before birth, well after the neuroepithelium and patterning have been generated from more GBC-like basal and apical progenitors. **Higher magnification staining confirming HBC markers present in regions containing neurogenic cells would be very helpful. The discrimination between bona fide olfactory HBCs and adjacent respiratory basal cells in scRNA-seq data sets containing both populations can be challenging, so careful annotation of these populations is important.** It is not clear that several of the markers selected here are well-established (COL17A1, RASSF6, NNAT), so some validation would be important. **The lncRNA MEG3 has been reported to be enriched in human olfactory HBCs compared to respiratory basal cells. Is this the case in these embryonic samples? If so, it may help to distinguish these populations.**

Response: We thank the reviewer for this thoughtful comment and agree that the developmental timing of olfactory horizontal basal cell (HBC) emergence is an important question, particularly in light of potential species-specific differences between human and mouse. Importantly, our revised data, shown in new Figure 2, do not indicate the presence of olfactory HBCs during the earliest stages of human OE development. However, both transcriptomic and spatial analyses consistently show that olfactory HBCs emerge later, at PCW10 and PCW12.

We selected a combination of canonical markers (e.g., KRT5, KRT17, TP63) together with additional candidate markers (COL17A1, RASSF6) to aid in annotation. The enrichment of MEG3 in olfactory HBCs provides further support for accurate discrimination. In our snRNA-seq, NNAT marker is clearly expressed in GBC. In addition, consistent with previous reports, MEG3 is enriched (wilcox-test, $p < 0.001$) in human olfactory HBCs compared to respiratory HBCs in these embryonic samples. This supports the molecular distinction of these populations.

These transcriptomic findings were independently corroborated by spatial validation. New immunofluorescence analyses at PCW7.5 and PCW9 showed a well-organized OE with neuronal and progenitor populations, but no detectable KRT5 expression within the OE at these stages; KRT5 immunoreactivity was restricted to the respiratory epithelium (RE), consistent with the absence of OHBCs during early development (Fig. 2i). By PCW12, KRT5-positive OHBCs were clearly present within the OE.

As suggested by the reviewer, Fig. 2j now provides high-magnification immunostaining for KRT5 and SOX2 at PCW12, clearly demonstrating KRT5-positive OHBCs localized above the basal lamina of the OE.

Together, these complementary approaches demonstrate that the emergence of OHBCs in the human OE occurs after initial neuroepithelial patterning and neurogenesis. We have revised the main text and the Discussion accordingly: *“Notably, whereas murine OHBCs are thought to emerge only shortly before birth, we demonstrate that OHBCs are present during early human OE development. This early emergence points to species-specific differences in basal cell deployment and suggests that HBCs may play broader developmental roles in humans beyond injury-induced regeneration”*.

Many of the canonical basic HLH TFs used to identify GBCs and INPs (ASCL1, NEUROG1, NEUROD1) seem to not be used here. Are these transcripts not present embryonically, or is there a reason they are not utilized?

Response: In our revised analysis, we re-annotated the neurogenic lineage following the reviewer’s suggestions, which clarified the progression from GBC → INP → iOSN. With this refinement, the canonical basic helix-loop-helix (bHLH) transcription factors are now clearly detected and appropriately used in our annotations.

Specifically, GBCs express HES6 together with ASCL1, consistent with early neurogenic priming reported previously (Durante et al., 2020). NEUROG1 and NEUROD1 are lowly expressed in INP, marking the transition toward committed neuronal precursors, while iOSNs express downstream neuronal differentiation markers such as GAP43, DCX, and TUBB3. Thus, these canonical bHLH TFs are indeed present embryonically in our dataset and are now incorporated directly into the annotation.

In addition, we observed that GBCs co-express several other bHLH family members, including TCF3, TCF4, TCF12, ARNT, and HIF1A, suggesting broader HLH involvement during progenitor regulation in the embryonic olfactory epithelium. This expanded set reinforces the developmental transitions captured in our updated annotation.

Several core analyses treat cells as the replicate rather than donors, which risks pseudoreplication and stage-donor confounding, especially at early stages where there is only a single donor (PCW7–9).

Response: we thank reviewer for pointing this out. To avoid pseudoreplication, all analyses of gene expression were performed using donor-level pseudobulk aggregation, summing counts per gene within each donor and cell type, and normalizing to logCPM prior to differential expression and visualization. As

shown in Fig. 2c and Extended Data Fig. 5b, cell type abundance was quantified using a donor-level pseudobulk approach. For each donor, we calculated the proportion of each cell type relative to the total cells from that donor, ensuring donors and not cells served as independent replicates. For statistical comparisons across developmental stages, we used an adaptive framework. When ≥ 3 donors were available per stage, we applied a linear mixed-effects model. For cell types with limited donor coverage (< 3 donors in any stage), we used a Kruskal–Wallis’s test, which is robust to small and unbalanced sample sizes. This approach minimizes sampling bias and accurately captures developmental changes in cell type composition.

The authors should describe how maternal contamination was assessed and excluded. At minimum, show that sex chromosome expression from the subjects is consistent with fetal sex (or inferred fetal sex), and that erythroid contamination is negligible. This is important because, while these early embryos will have immune cells of their own (particularly myeloid-lineage cells), some immune cells could be maternal.

Response: to address potential cross-sample contamination, we implemented a two-tiered quality control (QC) strategy evaluating (i) sex-based discordance and (ii) maternal erythroid ambient RNA. Sex QC followed the approach of Hao et al. (Cell, 2021), computing a \log_2 ratio of XIST to Y-chromosome gene expression (UTY, RPS4Y1, ZFY, DDX3Y, KDM5D) for each cell. Biological sex was inferred per sample by majority vote ($> 50\%$ of cells). In our dataset, male and female samples exhibited a clear bimodal distribution, with median sex scores of approximately +20 for female samples and -20 for male samples, corresponding to a ~ 41 -fold separation. Fewer than 0.5% of cells were classified as sex-discordant and were conservatively removed to minimize spurious cross-contamination signals.

Erythroid ambient RNA was quantified using canonical hemoglobin genes (HBB, HBA1/2, HBE1, HBG1/2, HBM). Given that human embryonic tissues at 7-12 PCW undergo extensive hepatic erythropoiesis, hemoglobin transcripts were broadly detectable (70-92% of cells), consistent with ambient RNA rather than true maternal contamination. To avoid over-filtering, removal was performed at the cluster level: true erythroid clusters were identified by co-expression of ALAS2, AHSP, HBB, HBA2, and GYPA, and only these clusters were excluded prior to downstream analysis. Thresholds were selected empirically from the data distribution and cross-validated against published embryonic atlases (Zhong et al., Nature 2018; Cao et al., Nature 2020). This conservative approach eliminated approximately 5% of cells while preserving genuine developmental populations.

The revised results section now reads as follows: “Sex was assigned using per-nucleus XIST-to Y-linked gene expression ratios, classifying nuclei as male-like, female-like, or ambiguous (Extended Data Fig. 3a,b). A small subset with discordant sex signatures and weak erythroid gene expression, consistent with minimal maternal blood-derived ambient RNA, was removed during quality control (Extended Data Fig. 3c,d). Low erythroid scores across samples ($\leq 2\%$) supported minimal contamination and yielded a sex- and donor-consistent dataset (Extended Data Fig. 3e).”

The authors state that Slingshot was run on UMAP coordinates. Because UMAP is a nonlinear visualization method and distances between cells are not preserved, it is not an appropriate space for trajectory inference. The analysis should be re-run on PCA, and the results plotted on UMAP only for display. This would ensure that inferred developmental trajectories reflect transcriptomic structure rather than visualization artifacts.

Response: we thank reviewer for this insightful comment. We agree that UMAP distortions can affect lineage inference and, in response, we have re-performed Slingshot entirely in PCA space, which preserves the global data manifold.

At PCW10 the authors highlight enrichment of hypoxia/stress/mitochondrial programs. These signatures are highly sensitive to cold ischemia and processing conditions in snRNA seq. The way to demonstrate biological, stage linked signal is to re test these enrichments using donor level statistics (pseudo bulk or

mixed effects), including stage as a fixed effect and donor as a random effect, and adding processing covariates where available (e.g., time to freezing). If the signal persists under those models, the claim is much stronger.

Response: we have re-evaluated the PCW10 hypoxia/stress/mitochondrial enrichments using donor-level pseudobulk analysis, aggregating counts by donor x cell type and modelling developmental stage as a fixed effect while accounting for donor using limma's blocking/duplicateCorrelation framework. Given the limited number of donors and the lack of detailed processing covariates (e.g., time to freezing), we consider the power of these models insufficient to support strong stage-linked claims. Accordingly, we have removed Fig. 2d and revised the Results to avoid over-interpretation.

Claims about changing cell type proportions (e.g., endothelial/lymphatic expansion, progenitor decline) likewise need donor aware compositional modeling to control for anatomical sampling.

Response: we agree with the reviewer that changes in cell-type proportions require donor-aware compositional analysis to control for anatomical sampling and unequal cell recovery. We therefore implemented a donor-level pseudobulk composition framework. For each donor d and cell type c , proportions were computed as:

$$\text{Proportion}(d,c) = \text{Cells}(d,c) / \sum_{c'} \text{Cells}(d,c'),$$

ensuring that each donor contributes a single, independent estimate of cell-type composition.

Donors were treated as biological replicates. Cell types represented in fewer than three donors per developmental stage were excluded from statistical comparison. Differences across stages were assessed using Kruskal–Wallis tests. To control for sequencing depth, mean UMI counts per cell type were residualized for PCW.

For cell–cell communication, the CellChat results probably reflect composition rather than biology. A more robust approach is to compute networks per donor (or balance donor groups), and report only ligand–receptor pathways that replicate across donors with FDR control and minimum cell count thresholds.

Response: we have removed the CellChat results, as proposed by reviewer 3.

For GO analysis, current best practices for snRNA seq favor performing enrichment on donor aware differential signals derived from a GLM (pseudo bulk) or mixed effects framework, rather than on pooled cell level lists. Suggest that you re run GO using such models and report FDR along with leading edge genes. The MERFISH panel included 57 ORs on coronal sections. The Discussion (line 647) states that there is evidence of a DV expression gradient or pattern of OR expression, yet the data do not clearly show this. Is it possible to better support this important finding.

Response: We thank the reviewer for raising this important point. In response, we now include in Extended Data Figure 11a (related to Spatial Transcriptomics data), a spatial map showing the total expression of the 57 ORs per cell, with only iOSNs highlighted. Although OSNs with high OR expression dominate the plot, the dorsoventral (DV) expression gradient described throughout the manuscript remains clearly visible.

Minor

In Seurat, `NormalizeData` uses `LogNormalize`, while `FindVariableFeatures(method="vst")` selects HVGs, but the current text conflates these.

Response: We thank the reviewer for noting this point. We have corrected the text to clearly distinguish the two steps: data were normalized using Seurat's `NormalizeData` with `LogNormalize`, and highly variable genes were identified using `FindVariableFeatures` with the "vst" method, following the standard Seurat workflow.

The cell cycle section needs reconciliation. The text says "non cycling cells were rare," yet the fraction of post mitotic cells rises from ~25% at PCW7 to >40% at PCW12, which is not rare. Consider updating the language to reflect the observed increase in post mitotic fractions with age.

Response: we thank reviewer for identifying this discrepancy. In the submitted version, G1-phase cells were incorrectly annotated as post-mitotic, for which we do apologize. This annotation error has now been corrected. In the revised analysis of the main olfactory epithelium, quantification across developmental time showed a progressive decline in actively cycling (G2/M) cells between PCW7 and PCW12 (~35% to ~16%), accompanied by an increase in G1 cells (~12% to ~40%), consistent with a transition from proliferation to differentiation (new Extended Data Fig. 6b). S-phase, post-mitotic, and non-cycling populations showed only modest temporal variation (new Extended Data Fig. 6b).

Reviewer #3:

This is a strong resource that will be useful to the community. I find the one neuron - one receptor observation particularly interesting; while consistent with mouse, establishing it directly in human fetal tissue is essential and, in my view, constitutes a discovery rather than mere validation, since this is the first systematic analysis in human.

Major comments:

One main criticism is that the manuscript is substantially over length for Nature Communications (limit ~5000 words for the main text; current draft ~7800) and the narrative loses focus as a result. In particular, the narration of GO terms from ED Fig 3 taking up the entire page 6 adds little, and the cell-cell communication panels in ED Fig 7, 8, 9 are not compelling without validation.

Walking through these figures panel-by-panel in the main text feels redundant and dilutes the story. I suggest substantially cutting the main text, removing scattered minor observations, and tightening to a few

unifying take-home insights. This would keep attention on the strongest contributions, namely the spatial organization of OE versus RE and the early emergence of predominantly single-OR expression in human.

Response: We agree with the reviewer's assessment. In response to this critique, we have substantially revised and streamlined the manuscript to improve focus and readability. Specifically, we removed the narration and presentation of the former Extended Data Figs. 3, 7, 8, and 9, which were considered insufficiently compelling and detracted from the central message. In addition, we eliminated several minor or tangential observations and condensed descriptive sections throughout the main text.

These revisions significantly reduce the overall length of the manuscript to better align with Nature Communications guidelines and sharpen the narrative around the core contributions of the study, namely the spatial organization of olfactory versus respiratory epithelium and the early emergence of predominantly single-odorant-receptor expression in human development.

Other specific concerns that the authors need to address in a revision include:

1. Provide experimental validation for the cell-cell communication analysis. At minimum, show in situ co-localization for some of the prioritized axis (for example SLIT-ROBO or SEMA3A-NRP1/PLXNA4).

Response: These data are no longer shown nor discussed in the revised manuscript.

2. The current approach for the GO analysis in ED Fig 3 is effectively pseudo-bulk gene expression across mixed cell types, so changes in cell type proportions likely drive the top terms rather than within-cell-type transcriptional changes. It would be more informative to perform differential analyses within stable cell types.

Response: ED Fig.3 has been removed from the revised manuscript.

3. ED Fig. 3c highlights hypoxia stress signal. Is this expected biologically, or could it reflect pre-analytical variables (reasons of pregnancy termination, ischemic interval, sample processing delay)?

Response: same as above.

4. The authors note a built-in CellPose segmentation for MERFISH but did not show how well it performed. Epithelia are very high density and hard to segment. Please include DAPI and a cell boundary marker overlaid with the final cell masks. This is essential, since precise segmentation underpins all single-cell analyses and is critical for evaluating the 1 neuron - 1 receptor principle.

Response: We agree with the reviewer that accurate cell segmentation is a critical early step in imaging-based spatial transcriptomics analyses, particularly in high-density epithelial tissues such as the olfactory epithelium. This consideration was a key reason for our use of the MERSCOPE platform, which incorporates dedicated cell boundary markers specifically to improve segmentation performance.

As an example, we have added in Extended Data Figure 11b, a representative figure showing DAPI and cell boundary staining overlaid with the final CellPose-derived segmentation masks for a representative region of section PCW9-P-S2. This visualization demonstrates the quality of nuclear and cellular segmentation achieved in dense epithelial regions.

In addition, for full transparency and reproducibility, we note that for each sample we are depositing a Xenium Explorer archive to GEO, which includes the raw staining images and the corresponding cell segmentation masks. This allows readers to directly inspect segmentation quality and supports the robustness of downstream single-cell analyses, including those related to the one-neuron-one-receptor principle.

5. In regulon results in Fig 4, several factors show a pronounced mismatch between expression and inferred activity (e.g., BARX2, NFKBIZ). How should this be interpreted

Response: We thank the reviewer for raising this important conceptual point. Transcription factor (TF) activity in our study was inferred from snRNA-seq data using the decoupleR package (v2.12.0) with a univariate linear model, which estimates TF activity based on the coordinated expression of downstream target genes rather than on the TF's own mRNA abundance. Curated TF-target regulatory interactions, weighted by mode of regulation, were used for this inference.

As a consequence, TF mRNA expression and inferred TF activity are not expected to be tightly correlated. Many TFs are primarily regulated at the protein level through post-translational mechanisms such as phosphorylation, degradation of inhibitory proteins, or nuclear translocation, none of which are captured by mRNA measurements. Accordingly, low TF mRNA abundance can coincide with high inferred activity if the TF protein is active and its downstream targets are transcriptionally upregulated.

This effect is particularly relevant for NF- κ B pathway components. For example, NFKBIZ (I κ B ζ) is an atypical I κ B family member that functions as an inducible co-activator of NF- κ B-dependent transcription. Its activity is highly stimulus-dependent and controlled at the protein level, including nuclear localization and interaction with other NF- κ B family members, which explains why inferred regulon activity can be high despite low or transient NFKBIZ mRNA expression.

Similarly, inferred activity of BARX2 without corresponding mRNA enrichment may reflect transient TF expression, persistence of downstream transcriptional effects, or partial motif and target overlap with related homeobox transcription factors (e.g., BARX1 or DLX family members). In such cases, decoupleR may attribute regulon activity to the most closely matching TF in the regulatory prior, even if another family member is the dominant active factor.

We have clarified this distinction between TF expression and inferred activity in the Methods section to guide interpretation of Fig. 4 and to avoid overinterpretation of TF mRNA levels as direct indicators of regulatory activity.

6. In Fig 5j-k, the curves for spatial distribution of cell types and gene expression are very hard to read. Is there a better way to plot this so the patterns are clearer?

Response: we have revised Fig.5j,k increasing the size of the gene expression labels. This should improve readability.

7. The dominance score in Fig 7 is a very good idea. But in Fig 7c-d the high-dominance OSNs look sparse. Is this very low abundance consistent with the scRNAseq single-OR rate? Could MERFISH sensitivity drop off and under-detect OR transcripts? Or could it be related to limitations of MERFISH gene panels only includes 57 OR genes?

Response: The reviewer is correct. We defined the dominance score to identify cells exhibiting a high degree of specificity in OR expression (cells expressing one OR while not expressing others). Our analysis showed that this strong specificity is characteristic of OSNs but is not present in all of them. As the reviewer suggests, there are several possible explanations for this observation:

- (i) We do not believe it results from limited MERFISH sensitivity leading to dropout or under-detection of OR transcripts, since in some OSNs we detected more than 200 molecules of the same OR showing their high degree of maturation;
- (ii) The inclusion of 57 ORs out of the approximately 400 total may represent the main source of apparent “leakiness”;
- (iii) Additionally, it is likely that at this developmental stage OSN maturation is still incomplete, which could also contribute to the variability observed.

8. The text references a Fig 7f panel that is not present. Is data omitted by mistake? Please restore the intended panel.

Response: This has been corrected in the revised version of the manuscript.

9. Beyond what is shown in Fig 7, it would be very interesting to look into whether individual ORs show spatial patterning. For example, specificity or gradients along the dorsal-ventral axis. Demonstrating this would showcase the unique advantages of spatial transcriptomics to reveal insights that other methods cannot, and would further strengthen the biological significance of the study.

Response: We thank the reviewer for this very interesting suggestion. We attempted to identify and extract spatial patterns of specific OR expression in iOSNs, as suggested; however, we found that the resulting trends were not sufficiently robust to support a clear conclusion. Nevertheless, we now include in Extended Data Figure 11c, a spatial representation of highly expressing iOSNs (i.e., cells in dominance categories above 3), colored by their dominant OR transcript.

Point-by-point response to reviewers' comments.

REVIEWERS' COMMENTS

Reviewer #2 (Remarks to the Author):

In general, the authors were highly responsive to my comments and those of the other Reviewers. As such, the revised manuscript is strengthened. Many of the problems from the original snRNA-seq analysis have now been corrected. The manuscript text has been edited to reduce lengthy descriptive commentary, improving readability. I have only minor concerns, not warranting re-review. Some of the conclusions on olfactory epithelial and olfactory receptor expression patterns from MERFISH remain overstated, because only 2 donor specimens are analyzed spatially (Extended Data Figure 11a) and only a small number of OR genes are visualized, with limited statistical analysis feasible, so this should be stated. This work should be a valuable resource and the authors should be commended for their efforts.

Reply: we have tempered our conclusions and included this sentence in the Discussion : “Importantly, spatial mapping also provided insight into the early establishment of OR expression patterns. Using MERFISH, we show that ORs are expressed in spatially restricted epithelial domains, suggesting that elements of the olfactory receptor map are established prenatally. However, because these analyses rely only on two donor specimens and a limited set of OR genes, these conclusions should be interpreted cautiously and validated in larger cohorts.”

Reviewer #3 (Remarks to the Author):

One remaining point is that Fig 5j remains difficult to interpret. While I understand that some improvement was made, the figure is still visually dense and the key spatial patterns are not easy to extract. I encourage the authors to consider an alternative visualization that makes the trends more immediately readable (for example, splitting into separate panels or using a heatmap or binned density representation). Overall, I feel the authors have done an appropriate revision and the manuscript is substantially strengthened.

Reply: We thank the reviewer for this helpful comment. We agree that the original presentation of Fig. 5j was visually dense and could be improved for clarity. In the revised version of the manuscript, we have updated Figure 5 accordingly by adopting a binned heatmap representation and reorganizing the panels to make the spatial trends more immediately interpretable. We believe this new visualization addresses the reviewer's concern and substantially improves readability.